# Non-stationary Extreme Value Analysis of Ground Snow Loads in the French Alps: a Comparison with Building Standards

Erwan Le Roux[1], Guillaume Evin[1], Nicolas Eckert[1], Juliette Blanchet[2], and Samuel Morin[3]

[1]Univ. Grenoble Alpes, INRAE, UR ETNA
[2]Univ. Grenoble Alpes, Grenoble INP, CNRS, IRD, IGE
[3]Univ. Grenoble Alpes, Univ. Toulouse, Météo-France, CNRS, CNRM, CEN Grenoble

**Correspondence:** Erwan Le Roux (erwan.le-roux@inrae.fr)

**Abstract.** In a context of climate change, trends in extreme snow loads need to be determined to minimize the risk of structure collapse. We study trends in 50-year return levels of ground snow load (GSL) using non-stationary extreme value models. These trends are assessed at a mountain massif scale from GSL data, provided for the French Alps from 1959 to 2019 by a meteorological reanalysis and a snowpack model. Our results indicate a temporal decrease in 50-year return levels from 900 m to 4200 m, significant in the Northwest of the French Alps until 2100 m. We detect the most important decrease at 900 m with an average of $-30\%$ for return levels between 1960 and 2010. Despite these decreases, in 2019 return levels still exceed return levels designed for French building standards under a stationary assumption. At worst, i.e. at 1800 m, return levels exceed standards by $15\%$ on average, and half of the massifs exceeds standards. We believe that these exceedances are due to questionable assumptions concerning the computation of standards. For example, these were devised with GSL, estimated from snow depth maxima and constant snow density set to $150 \text{ kg m}^{-3}$, which underestimate typical GSL values for the snowpack.

## 1 Introduction

Extreme snow loads can generate economic damages and casualties. For instance, more than \$200 million in roof damages occurred during the Great Blizzard of 1993 (O'Rourke and Auren, 1997). In 2006, at the Katowice International Fair, the roof of one of the buildings collapsed under a layer of snow, leading to 65 casualties and 140 injured (BBC News, 2006). In France, snow loads over Roussillon in 1986, caused both 17 million euros in damages and a major power outage due to overloading of electrical cables and pylons by sticking snow (Vigneau, 1987; Naaim-Bouvet et al., 2000).

Ground snow load (GSL) is defined as the pressure exerted by accumulated snow on the ground, which can be directly associated with accumulated snow on structures, e.g. on roofs (Sanpaolesi et al., 1998). In details, the observed height of accumulated snow is called snow depth (in m). The density of this snow can vary widely between precipitation particles ($\rho_{\text{SNOW}} \approx 100 \text{ kg m}^{-3}$) and a ripe snowpack ($\rho_{\text{SNOW}} \approx 500 \text{ kg m}^{-3}$). Multiplying snow depth by snow density gives the surfacic mass of snow (in kg m$^{-2}$). Surfacic mass of snow corresponds to the snow water equivalent (SWE) which is the height of water (in mm) we could obtain if we melt all the snow in a 1 m$^2$ area. Indeed, since water density is $\rho_{\text{WATER}} = 1000 \text{ kg m}^{-3}$, we have that 1 mm of water on 1 m$^2$ has a surfacic mass of $1 \text{ kg m}^{-2}$. Snow load is the pressure exerted by this surfacic mass of snow (in N m$^{-2}$ or Pa) and equals the SWE times the gravitational acceleration (g = 9.81 m s$^{-2}$).

Snowpack variables related to GSL (snow depth, SWE) evolve with climate change. As shown in Table 1, literature on past trends in snowpack variables for the Western Alps shows a decreasing trend. Literature on projected trends also points to a decrease (stronger for the second half of the 21st century under a high greenhouse gas emission scenario than with strong reductions in greenhouse gas emissions) for mean winter (December-May) SWE in the European Alps (IPCC, 2019). However, anthropogenic climate change impacts climatic variables in their averages, but also in their extremes (Klein Tank and Können, 2003; IPCC, 2012). For instance, annual maxima of snow depth have decreased in Switzerland (Marty and Blanchet, 2012).

Projected trends in extreme snowpack variables are prone to strong uncertainties (Strasser, 2008; Beniston et al., 2018) as both mean winter temperature (IPCC, 2019) and winter precipitation extremes (Rajczak and Schär, 2017) are projected to increase in the European Alps.

| Variable | Indicator | Trend | Country | Time | Source |
|---|---|---|---|---|---|
| HS | Seasonal mean (Nov to Apr) | Decrease | CH | 1931-1999 | Laternser and Schneebeli (2003) |
| | Winter mean (Dec to Feb) | Decrease in the North | FR | 1958-2007 | Durand et al. (2009b) |
| | Mean annual maxima | Decrease | CH | 1930-2010 | Marty and Blanchet (2012) |
| | Seasonal mean (Nov to May) | Decrease | IT | 1951-2010 | Terzago et al. (2013) |
| | Seasonal mean (Nov to Apr) | Decrease in the South | CH | 1961–2012 | Schöner et al. (2019) |
| SWE | 1st of April value | Decrease | IT | 1965-2007 | Bocchiola and Diolaiuti (2010) |
| | 1st of April value | Decrease | FR, IT, CH | 1968-2012 | Marty et al. (2017) |

**Table 1.** Past trends in snowpack variables, snow depth (HS) and snow water equivalent (SWE), according to existing studies in the Western Alps, i.e. in Italy (IT), France (FR), and Switzerland (CH). In the Trend column, "North" and "South" refer to the considered country.

The impact of climate change on GSL was not taken into account in current European standards for structural design, a.k.a

Eurocodes (Sanpaolesi et al., 1998), which drive French standards (Biétry, 2005). These standards define that structures must withstand their own weight plus a pressure proportional to a characteristic value. The latter is the stationary 50-year return level of GSL, exceeded once every 50 years on average. Thus, studying trends in 50-year return levels of GSL is needed for updating these standards (Croce et al., 2018). In the literature, past and projected trends in 50-year return levels of GSL have rarely been investigated with the exception of (Rózsás et al., 2016; Il Jeong and Sushama, 2018; Croce et al., 2018). In the French

Alps, several studies focused on extreme snow variables (Biétry, 2005; Gaume et al., 2012, 2013) and their spatial dependence (Nicolet et al., 2015, 2016, 2017, 2018). However, trends in 50-year return levels of GSL remain unexplored.

We fill these gaps by studying annual maxima of GSL provided every 300 m of altitude at a mountain massif scale for the 23 French Alps massifs. We rely on the SAFRAN-Crocus reanalysis (Vernay et al., 2019) produced by the SAFRAN-Crocus chain (Durand et al., 2009a; Vionnet et al., 2012) available for the period 1959-2019. The major advantage of this reanalysis

is to take benefit of an advanced snowpack model which provides daily estimates of ground snow load values, while previous studies relied on approximate values directly related to snow depth with a crude estimation of snow density (Biétry, 2005). Thus, our approach considers only natural snow processes, i.e. we do not account for snow removal throughout the year and consider all processes (accumulation, thaw/freeze, melt, compaction etc.) occurring during the winter season.

Our statistical methodology consists in applying stationary and non-stationary extreme value models to annual maxima time series. We select one model by massif and altitude with the AIC statistical criterion, validate the selected model with the Anderson-Darling test, and assess its significance with the likelihood ratio statistical test. Finally, for each massif and altitude, we compute the relative change of 50-year return levels of GSL between 1960 and 2010, and we compare the non-stationary return level in 2019 with the stationary return level designed for French building standards.

This paper is organized as follows. Section 2 presents our data. Section 3 describes standards for ground snow load. Then, section 4 explains our methodology. Results, discussion and conclusions are introduced in Sections 5, 6 and 7, respectively.

## 2 Ground snow load data

The study area covers the French Alps which are located between Lake Geneva to the north and the Mediterranean Sea to the south (Fig. 1). The climate is contrasted, colder and wetter in the northern Alps and much drier in the southern Alps (Durand et al., 2009a). This region is typically divided into 23 mountain massifs of about 1000 km$^2$. We rely on the SAFRAN-Crocus reanalysis (Vernay et al., 2019) from the SAFRAN-Crocus chain (Durand et al., 2009a; Vionnet et al., 2012) available from August 1958 to July 2019 at the scale of these massifs, every 300 m of altitude from 300 m to 4800 m. Contrary to gridded products, this reanalysis assumes for a given altitude the homogeneity of the different variables at the scale of the massif. Also, annual maxima are available from 1959 to 2019. Indeed, annual maxima denote the maxima during a year centered on the winter season, e.g. annual maxima for 1959 correspond to the maxima from the 1st of August 1958 to the 31st of July 1959.

To sum up, GSL equals SWE from the SAFRAN-Crocus reanalysis times the gravitational acceleration. We study time series of annual maxima of GSL for each massif from 1959 to 2019 every 300 m of altitude from 300 m to 4800 m (Fig. 1).

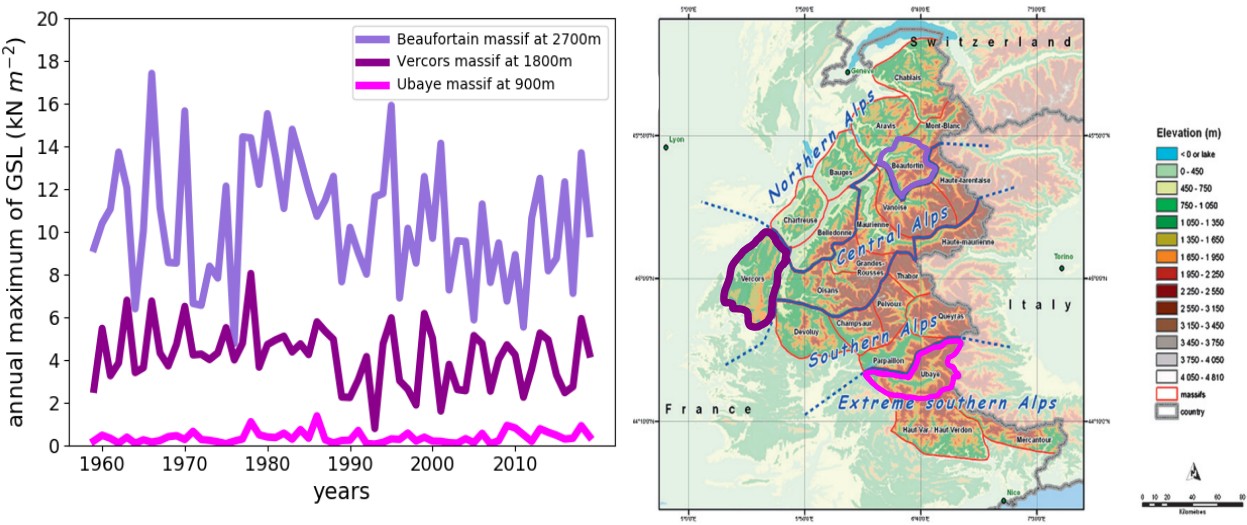

**Figure 1.** Left: Three time series of annual maxima of ground snow load (GSL) from 1959 to 2019 for 3 massifs at low (900 m), mid (1800 m) or high (2700 m) altitude. Right: 23 mountains massifs of the French Alps and their orographic features (Durand et al., 2009a).

The SAFRAN-Crocus reanalysis is produced by a chain of two models. First, SAFRAN meteorological reanalysis (Durand et al., 2009a) performs a spatialization of the weather data (precipitation, temperature, humidity, radiation, wind speed) over the massifs and altitudes. Then, the Crocus snowpack model (Vionnet et al., 2012) infers snow depth and SWE based on SAFRAN time series. Crocus is a one-dimensional multilayer physical snow scheme, which simulates the snowpack evolution over time, by accounting for several processes such as thermal diffusion, phase changes and metamorphism.

The SAFRAN-Crocus reanalysis has been evaluated against various observation datasets, as reported in previous publications (Lafaysse et al., 2013; Vionnet et al., 2016; Revuelto et al., 2018; Vionnet et al., 2019). In most cases, the evaluation is carried out against in-situ snow depth observations and remote sensing snow cover information. For example, Vionnet et al. (2016) evaluated SAFRAN-Crocus snow depth data against 79 observed snow depth data in the French Alps for the 2010-2014 time period, with mean bias and standard error values of 18 cm and 37 cm, respectively. This corresponds to typical values for snow modelling systems applied in various regions on Earth. Because of lower data availability, evaluations against observed SWE values are less frequent than against snow depth data, although we note that Crocus has been shown to perform extremely well compared to other snow cover models, in terms of SWE, across many observation sites worldwide (Krinner et al., 2018) and SAFRAN-Crocus exhibits satisfying performance in terms of snow depth and SWE in the Pyrenees (Quéno et al., 2016), providing confidence, with respect to other existing datasets, in using this model chain for GSL values. Further model evaluations, using additional datasets, are required to continue assessing and improving the quality of the model chain. Furthermore, we highlight that we only used SAFRAN-Crocus reanalysis values on flat field, and we did not used simulations on slopes, hence it is not relevant to discuss the impact of slope and aspect on the results of this study.

## 3 Standards for ground snow load in the French Alps

GSL French standards (Biétry, 2005) are based mostly on Eurocodes (Sanpaolesi et al., 1998) and on prior French standards (AFNOR, 1996). Each French department, and by extension each French Alps massif, is associated with a region (C or E) that sets characteristic 50-year return level values of GSL between 200 m and 2000 m of altitude (Fig. 2).

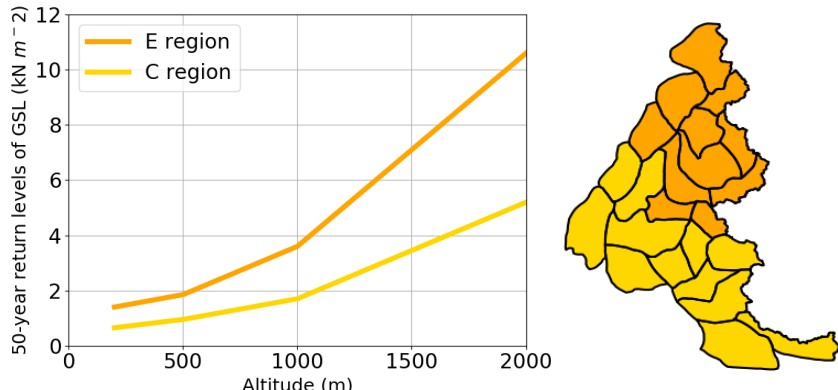

**Figure 2.** Left: French standards 50-year return levels of ground snow load (GSL) w.r.t. altitude for regions C and E. Right: Regions map.

French standards were elaborated with annual maxima time series of snow depth on the ground measured at stations from 1945 to 1992. GSL data were approximated from annual maxima of snow depth and by assuming that snow density equals $150 \, \text{kg m}^{-3}$. Following Eurocodes, the characteristic value of GSL is defined as the 50-year return level of a Gumbel distribution (Sect. 4). This distribution was fitted using the least squares method and by removing the top annual maximum when considered exceptional (Biétry, 2005) according to a criterion not explicitly mentioned in the French report cited as reference. However, in the Eurocodes, the standard method was to consider the top maximum as exceptional if it was 1.5 times larger than the second largest maximum (Sanpaolesi et al., 1998). In our methodology, we do not remove the top annual maximum.

## 4 Statistical Methodology

Following extreme value theory, we employ 2 stationary models and 6 non-stationary models for time series of annual maxima of GSL (Sect. 4.1). We select a single model for each time series (i.e. for each massif and altitude) with the AIC statistical criterion, validate this model with the Anderson-Darling test, and assess its significance with the likelihood ratio statistical test (Sect. 4.2). Finally, we compute the relative change of 50-year return levels of GSL between 1960 and 2010, quantify the uncertainty of return levels in 2019 to compare them with the stationary return levels designed for French standards (Sect. 4.3).

### 4.1 Stationary and non-stationary models based on extreme value distributions

Climate extremes are generally studied with statistics. As underlined in the IPCC special report on climate extremes, a large amount of statistical literature builds on extreme indices to examine moderate extremes (IPCC, 2012). However, since we focus on extremes that are more rare, it is recommended to rely on extreme value theory (EVT, Coles 2001). Such statistical models provide and hypothesize additional prior information in order to compensate the limited amount of empirical observations that commonly span only several decades. These models can be used to extrapolate beyond the empirical observations, and to estimate return levels (Sect. 4.3).

EVT offers a suitable framework to analyse extreme values, i.e. to model the form of the tail for almost any probability distribution. Asymptotically, as the central limit theorem motivates sample means modelling with the normal distribution, the Fisher–Tippett–Gnedenko theorem (Fisher and Tippett, 1928; Gnedenko, 1943) encourages sample maxima modelling with the GEV distribution. This theorem justifies that the maximum of finite-sized blocks with a large enough block size can be modeled with the GEV distribution. In practice, an annual maximum is thus usually considered as a realization of a GEV distribution. Three parameters define the GEV distribution: a location $\mu$, a scale $\sigma > 0$ and a shape $\zeta$ (a.k.a extremal index or tail index). The GEV distribution includes three specific types of distributions: Weibull ($\zeta < 0$), Fréchet ($\zeta > 0$) and Gumbel ($\zeta = 0$). Thus, by definition, if $Z$ represents an annual maximum of GSL, we can assume that $Z$ follows a GEV distribution, i.e. $Z \sim \text{GEV}(\mu, \sigma, \zeta)$, which implies that:

$$P(Z \leq z) = \begin{cases} \exp\left[-\left(1 + \zeta \frac{z-\mu}{\sigma}\right)_+^{-\frac{1}{\zeta}}\right] & \text{if } \zeta \neq 0 \text{ and where } u_+ \text{ denotes } \max(u, 0), \\ \exp\left[-\exp\left(-\frac{z-\mu}{\sigma}\right)\right] & \text{if } \zeta = 0, \text{ in other words if } Z \sim \text{Gumbel}(\mu, \sigma). \end{cases} \tag{1}$$

In a context of climate change, a large amount of hydrological literature builds on non-stationary modelling (Milly et al., 2008) to assess whether a time series is generated by a unique probability distribution (stationary model), or if the generating probability distribution is changing (non-stationary model). Non-stationary extremes are usually studied with both non-stationary modelling and EVT (Katz et al., 2002). Annual maxima are assumed independent but not necessarily identically distributed (Serinaldi and Kilsby, 2015). Such approaches combine a stationary random component (a fixed extreme value distribution) with non-stationary deterministic functions that map each temporal covariate $t$ to the changing parameters of the distribution (Montanari and Koutsoyiannis, 2014). In a non-stationary context, Zhang et al. (2004) showed that tests based on this parametric approach have stronger power of detection when compared with non-parametric methods.

We consider non-stationarity for both the Gumbel distribution and the more general GEV distribution, since they represent natural extensions of the Gumbel distribution which was used for French building standards (Sect. 3). For any model, we have $Z(t) \sim \text{GEV}(\mu(t), \sigma(t), \zeta(t))$, as the Gumbel distribution corresponds to $\zeta(t) = 0$. For a model $\mathcal{M}$, we denote as $\boldsymbol{\theta}_{\mathcal{M}}$ all parameters for its functions ($\mu(t)$, $\sigma(t)$ and $\zeta(t)$). We focus on simple linear functions due to the limited length of time series (60 years). The linearity starts in 1959 which is the first winter with available data. As shown in Table 2, we consider only models with a constant shape parameter, but where the location and/or the scale parameter can vary linearly with years $t$.

| Model type | Distribution | Model name | $\mu(t)$ | $\sigma(t)$ | $\zeta(t)$ | $\boldsymbol{\theta}_{\mathcal{M}}$ | # $\boldsymbol{\theta}_{\mathcal{M}}$ |
|---|---|---|---|---|---|---|---|
| Stationary | Gumbel | $\mathcal{M}_0$ | $\mu_0$ | $\sigma_0$ | 0 | $(\mu_0, \sigma_0)$ | 2 |
| | GEV | $\mathcal{M}_{\zeta_0}$ | | | $\zeta_0$ | $(\mu_0, \sigma_0, \zeta_0)$ | 3 |
| Non-stationary | Gumbel | $\mathcal{M}_{\mu_1}$ | $\mu_0 + \mu_1 \times (t - 1959)$ | $\sigma_0$ | 0 | $(\mu_0, \mu_1, \sigma_0)$ | 3 |
| | GEV | $\mathcal{M}_{\zeta_0, \mu_1}$ | | | $\zeta_0$ | $(\mu_0, \mu_1, \sigma_0, \zeta_0)$ | 4 |
| Non-stationary | Gumbel | $\mathcal{M}_{\sigma_1}$ | $\mu_0$ | $\sigma_0 + \sigma_1 \times (t - 1959)$ | 0 | $(\mu_0, \sigma_0, \sigma_1)$ | 3 |
| | GEV | $\mathcal{M}_{\zeta_0, \sigma_1}$ | | | $\zeta_0$ | $(\mu_0, \sigma_0, \sigma_1, \zeta_0)$ | 4 |
| Non-stationary | Gumbel | $\mathcal{M}_{\mu_1, \sigma_1}$ | $\mu_0 + \mu_1 \times (t - 1959)$ | $\sigma_0 + \sigma_1 \times (t - 1959)$ | 0 | $(\mu_0, \mu_1, \sigma_0, \sigma_1)$ | 4 |
| | GEV | $\mathcal{M}_{\zeta_0, \mu_1, \sigma_1}$ | | | $\zeta_0$ | $(\mu_0, \mu_1, \sigma_0, \sigma_1, \zeta_0)$ | 5 |

**Table 2.** Statistical models considered for annual maxima of GSL are based on the Gumbel or the GEV distribution, and are extensions of the stationary Gumbel model. For non-stationary models, the location and/or the scale vary linearly with years $t$ after the starting year 1959.

## 4.2 Model selection, validation and significance

**Model selection**. Let $\boldsymbol{z} = (z_{1959}, ..., z_{2019})$ represent a time series of annual maxima of GSL, i.e. for a massif and an altitude (Sect. 2). First, models are fitted with the maximum likelihood method. For every model $\mathcal{M}$, we compute the maximum likelihood estimator $\widehat{\boldsymbol{\theta}}_{\mathcal{M}}$ which corresponds to the parameter $\boldsymbol{\theta}_{\mathcal{M}}$ that maximizes the likelihood:

$$\widehat{\boldsymbol{\theta}}_{\mathcal{M}} = \underset{\boldsymbol{\theta}_{\mathcal{M}}}{\text{argmax}} \, \mathcal{L}(\boldsymbol{\theta}_{\mathcal{M}}; \boldsymbol{z}) \text{ where } \mathcal{L}(\boldsymbol{\theta}_{\mathcal{M}}; \boldsymbol{z}) = p(\boldsymbol{z}|\boldsymbol{\theta}_{\mathcal{M}}) = \prod_t p(z_t|\mu(t), \sigma(t), \zeta(t)) = \prod_t \frac{\partial P(Z(t) \leq z_t)}{\partial z_t}. \tag{2}$$

Then, for each $z$, i.e. for each massif and altitude, we select the model $\mathcal{M}_N$ with the minimal AIC value (Akaike, 1974), as it is the best information criterion in a non-stationary context with small sample sizes (Kim et al., 2017). We define:

$$\mathcal{M}_N = \underset{\mathcal{M} \in \text{Table 2}}{\mathrm{argmin}}\ \mathrm{AIC}(\mathcal{M}) \text{ where } \mathrm{AIC}(\mathcal{M}) = 2 \times [\#\boldsymbol{\theta}_{\mathcal{M}} - \log \mathcal{L}(\widehat{\boldsymbol{\theta}}_{\mathcal{M}}; z)], \text{ where } \#\boldsymbol{\theta}_{\mathcal{M}} \text{ is the cardinality of } \boldsymbol{\theta}_{\mathcal{M}}. \tag{3}$$

The selected model $\mathcal{M}_N$ can be any model from Table 2, i.e. a stationary or a non-stationary model. The subscript $N$ designates the number of additional parameters compared to the stationary Gumbel model $\mathcal{M}_0$, i.e. $N = \#\boldsymbol{\theta}_{\mathcal{M}_N} - \#\boldsymbol{\theta}_{\mathcal{M}_0}$.

**Model validation**. Quantile-Quantile (Q-Q) analysis is performed for all selected models. To apply this analysis to both stationary and non-stationary model, we rely on Richard W. Katz (2012) that suggests (i) to transform the data to stationary Gumbel (ii) to use a Q-Q plot analysis on the transformed data w.r.t. to a Gumbel distribution. Q-Q plots reveal that transformed data is well fitted by a stationary Gumbel distribution, hence that data is well fitted by the selected models (App. B). Moreover, according to the comparative study of Abidin et al. (2012), the most powerful Goodness of Fit test for the Gumbel distribution is a combination of the Anderson-Darling test and the Maximum Likelihood Estimator. We apply this test on the transformed data using Saeb (2018), and found that we cannot reject the null hypothesis (samples generated from the Gumbel model) at the 5% significance level for almost all our selected models (98%), justifying their good fit. We refer to App. B for more details.

**Model significance**. If the selected model $\mathcal{M}_N$ is not the model $\mathcal{M}_0$ then, since models are nested, we can compute the significance of $\mathcal{M}_N$ w.r.t. $\mathcal{M}_0$ with a likelihood ratio test (Coles, 2001). This test assess whether there is enough evidence to reject the stationary Gumbel model $\mathcal{M}_0$ in favor of the selected model $\mathcal{M}_N$. The null hypothesis can be stated as: the $N$ additional parameters of the model $\mathcal{M}_N$ can be set to zero. In other words, we want to check if setting to zero the $N$ additional parameters of the model $\mathcal{M}_N$ is supported by the data $z$. Under the null hypothesis, the likelihood ratio test statistic (LR) has an asymptotic $\chi_N^2$-distribution: $\mathrm{LR}(\widehat{\boldsymbol{\theta}}_{\mathcal{M}_N}, \widehat{\boldsymbol{\theta}}_{\mathcal{M}_0}, z) = -2\log\frac{\mathcal{L}(\widehat{\boldsymbol{\theta}}_{\mathcal{M}_0}; z)}{\mathcal{L}(\widehat{\boldsymbol{\theta}}_{\mathcal{M}_N}; z)} \dot{\sim} \chi_N^2$, where $\dot{\sim}$ means distributed under suitable regularity conditions. In practice, the test works as follows. We first choose a $0.05$ level of significance. Then, if LR is greater than $q_{\chi_N^2}$, the $1 - 0.05 = 0.95$ quantile of the $\chi_N^2$ distribution, we reject the nested model $\mathcal{M}_0$ in favor of the selected model $\mathcal{M}_N$. If the selected model $\mathcal{M}_N$ is non-stationary, we consider the associated trend as significant.

## 4.3 Return levels

In a stationary context, the $T$-year return level, which corresponds to a return period of $T$ years, is the classical metric to quantify hazards of extreme events (Cooley, 2012). For a stationary model, there is a one-to-one relationship between a return level (a quantile exceeded each year with probability $p$) and a return period (a duration exceeded every $T = \frac{1}{p}$ years on average).

In a non-stationary context, return level and return period concepts (Cooley, 2012) become further ambiguous, prone to misconceptions and can lead to misleading conclusions (Serinaldi, 2015). We focus on the yearly level for a fixed probability of exceedance, a.k.a effective return level (Katz et al., 2002; Cheng et al., 2014), as it conveys best that hazard evolves with time.

For the stationary Gumbel model $\mathcal{M}_0$, the return level $z_p(\boldsymbol{\theta}_{\mathcal{M_0}})$ is defined as the level exceeded each year with probability $p$. In other words, if $Z$ denotes an annual maximum, then $P(Z \leq z_p(\boldsymbol{\theta}_{\mathcal{M_0}})) = 1 - p$. This return level is constant through time and equals $z_p(\boldsymbol{\theta}_{\mathcal{M_0}}) = \mu_0 - \sigma_0 \log(-\log(1-p))$. In this paper, we set $p = \frac{1}{50} = 0.02$ as it corresponds to the 50-year return period defined by French standards (based on European standards) for the design working life of buildings (Sect. 3).

     For the selected model $\mathcal{M}_N$, the return level is defined as the yearly level for a fixed probability of exceedance $p$. For any

model considered in Table 2, we obtain $z_p(\boldsymbol{\theta}_{\mathcal{M_N}}, t) = \mu_0 + \mu_1 \times (t - 1959) - \frac{\sigma_0 + \sigma_1 \times (t - 1959)}{\zeta_0}[1 - (-\log(1-p))^{-\zeta_0}]$, where we set $\mu_1, \sigma_1$ or $\zeta_0$ to 0 if they are not defined in the model $\mathcal{M}_N$. For example, for the Gumbel model $\mathcal{M}_0$, the return level is constant: for any year $t$, $z_p(\boldsymbol{\theta}_{\mathcal{M_0}}, t) = \lim_{\zeta_0 \to 0}[\mu_0 + \frac{\sigma_0}{\zeta_0}(1 - (-\log(1-p))^{-\zeta_0})] = \mu_0 - \sigma_0 \log(-\log(1-p))$.

     For any considered model, the time derivative of the return level is constant, as $\frac{\partial z_p(\boldsymbol{\theta}_{\mathcal{M_N}}, t)}{\partial t} = \mu_1 - \frac{\sigma_1}{\zeta_0}(1 - (-\log(1-p))^{-\zeta_0})$. It quantifies the yearly change of return level. Thus, the relative difference of return levels between year $t_1$ and year $t_2$ is:

Relative change$(z_p(\boldsymbol{\theta}_{\mathcal{M_N}}, t_1), z_p(\boldsymbol{\theta}_{\mathcal{M_N}}, t_2)) = \dfrac{z_p(\boldsymbol{\theta}_{\mathcal{M_N}}, t_2) - z_p(\boldsymbol{\theta}_{\mathcal{M_N}}, t_1)}{z_p(\boldsymbol{\theta}_{\mathcal{M_N}}, t_1)} = \dfrac{t_2 - t_1}{z_p(\boldsymbol{\theta}_{\mathcal{M_N}}, t_1)} \times \dfrac{\partial z_p(\boldsymbol{\theta}_{\mathcal{M_N}}, t)}{\partial t}. \quad (4)$

     In the context of maximum likelihood estimation, uncertainty related to return levels can be derived by the delta method, which quickly provides confidence intervals both in the stationary and non-stationary case (Coles, 2001; Gilleland and Katz, 2016). First, the return level estimator associated with the maximum likelihood estimator simply equals $z_p(\widehat{\boldsymbol{\theta}}_{\mathcal{M}})$. Then, due to the asymptotic normality of the maximum likelihood estimator (MLE), we can assume that, even with a finite number of

185 data, the MLE is normally distributed. Therefore, under regularity conditions, limits of the $1 - \alpha = 95\%$ confidence interval are $\widehat{\boldsymbol{\theta}}_{\mathcal{M}} \pm q_{\frac{\alpha}{2}} \times v_{z_p}(\widehat{\boldsymbol{\theta}}_{\mathcal{M}})$ where $q_{\frac{\alpha}{2}}$ is the 1-$\frac{\alpha}{2}$ quantile of the standard normal distribution, and $v_{z_p}$ is a function that associates to each parameter $\boldsymbol{\theta}_{\mathcal{M}}$ the variance of the approximate normal distribution associated with its return level $z_p(\boldsymbol{\theta}_{\mathcal{M}})$. For a full expression of the function $v_{z_p}$ and for details on the delta method, we refer to Theorem 2.4 of Coles (2001). In particular, this theorem explains that the delta method is valid for $\zeta_0 < 1$, which is respected in our case as $-0.5 \leq \zeta_0 \leq 0.5$ (Sect. 4.4). Also,

uncertainty of non-stationary return levels $z_p(\widehat{\boldsymbol{\theta}}_{\mathcal{M}}, t)$ can be obtained by incorporating the covariate $t$ in the function $z_p$.

## 4.4   Application

     First, we exclude 4 times series of annual maxima with more than $10\%$ of zeros, i.e. years without GSL. Then, we fit models to time series, and retain only those models with $-0.5 \leq \widehat{\zeta}_0 \leq 0.5$. This impacts 3 time series. We make this choice because $\widehat{\zeta}_0 > 0.5$ designates distributions with an "exploding" tail which are known to be physically implausible (Martins and Stedinger,

2000). Following Sect. 4.2, we select one model for each time series (i.e. for each massif and altitude) with the AIC statistical criterion. Then, we exclude the 5 times series ($2\%$) where the selected model do not pass the Anderson test. Finally, we assess if the selected model is significantly more appropriate than the stationary Gumbel model $\mathcal{M}_0$ with a likelihood ratio test.

## 5 Results

### 5.1 Selected models

Figure 3 shows that a stationary model, i.e. models $\mathcal{M}_0$ and $\mathcal{M}_{\zeta_0}$, is selected for a majority (57%) of time series studied (Sect. 2). Models with a linearity in both the location and scale parameters are the most frequently selected non-stationary models (22%). For both stationary and non-stationary models, Gumbel models are always more often selected that their corresponding GEV models (Fig. 3, Fig. 4). All in all, we highlight that 39% of selected models are significantly more appropriate than the stationary Gumbel model $\mathcal{M}_0$.

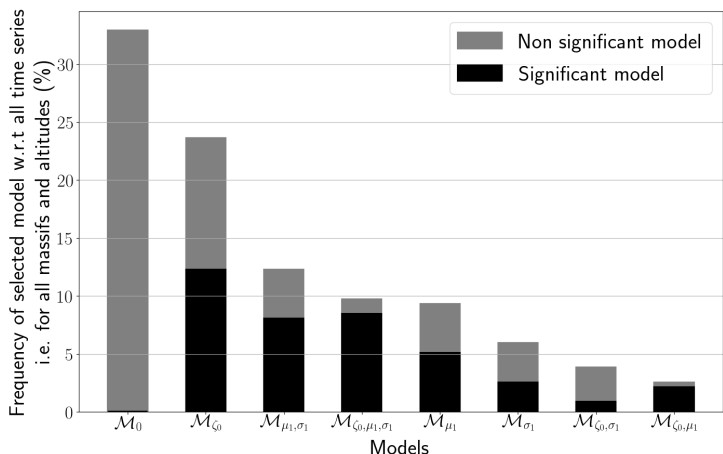

**Figure 3.** Distribution of selected models. Frequency of selected model (in %) w.r.t. all time series, i.e. for all massifs and altitudes. For the selection procedure and the definition of significance, we refer to Sect. 4.2.

Figure 4 depicts shape parameter values for the selected models at 900 m, 1800 m and 2700 m. We notice that a majority of massifs are white illustrating that a (stationary or non-stationary) Gumbel model (i.e. $\widehat{\zeta}_0 = 0$) is selected (Sect. 5). This emphasizes that a Gumbel distribution often explains more succinctly the data than a GEV distribution. Also, with the GEV distribution, the estimated most likely shape parameter $\widehat{\zeta}_0$ is often quite uncertain, i.e. confidence intervals are large, which is the main reason why French standards did not rely on it. This uncertainty in $\widehat{\zeta}_0$ likely comes from the limited length of time series. Therefore, additional data would enable to estimate $\widehat{\zeta}_0$ more robustly, and thus reduce uncertainty.

In Figure 4, we further observe that non-null shape parameters at low altitudes (900 m) are always positive (brown-colored massifs), i.e. a Fréchet distribution is preferred. On the other hand, for high altitudes (1800 m and 2700 m) non-null shape parameters are always negative (green-colored massifs), i.e. a Weibull distribution is favored. Similar results for the shape parameter have been observed for snow depth by Blanchet et al. (2009), Blanchet and Lehning (2010) and Schellander and Hell (2018). This reflects the different nature of annual maxima of GSL between low and high altitudes. At high altitudes, annual maxima are mainly due to snowpack accumulation during several months, while at low altitudes this accumulation is limited, and thus annual maxima roughly correspond to heavy precipitation.

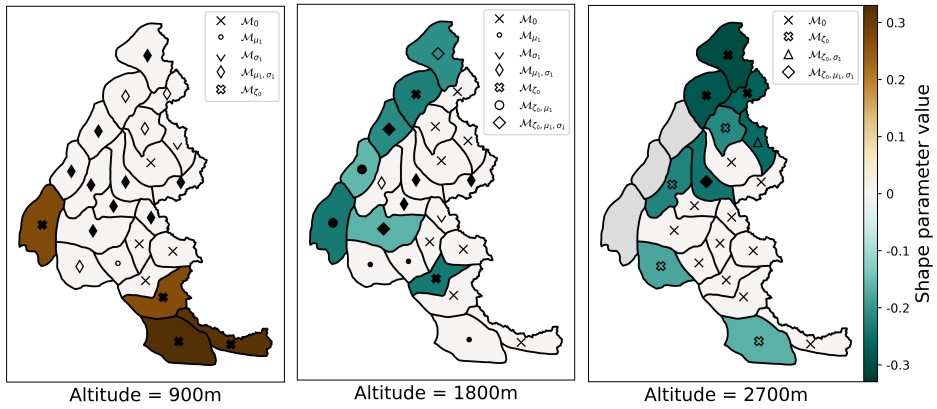

**Figure 4.** Shape parameter values for the selected models at low (900 m), mid (1800 m) or high (2700 m) altitude. Markers show selected model $\mathcal{M}_N$ while filled markers symbolize models that are significantly better than the Gumbel model $\mathcal{M}_0$ (Sect. 4.2). Grey areas denote either time series that were excluded (Sect. 4.4) or missing data, e.g. when the altitude considered is above the top altitude of the massif.

## 5.2 Trends in return levels of ground snow load

Figure 5 maps the relative change of 50-year return levels of GSL between 1960 and 2010 (Eq. 4) at 900 m, 1800 m and 2700 m (see App. A for maps at all altitudes). Quantitatively, for Northwest massifs, we observe that return levels have decreased by up to $60\%$ at 900 m (dark blue), while at 1800 m this decrease is less marked (pale blue). Qualitatively, these decreasing trends are frequently due to significant changes both in the location and scale parameters of the Gumbel or GEV distribution (small and large diamond-shaped filled markers). At 2700 m, or in the South at 900 m and 1800 m, we often do not observe any trends (white), since stationary models are selected (small and large cross-shaped markers).

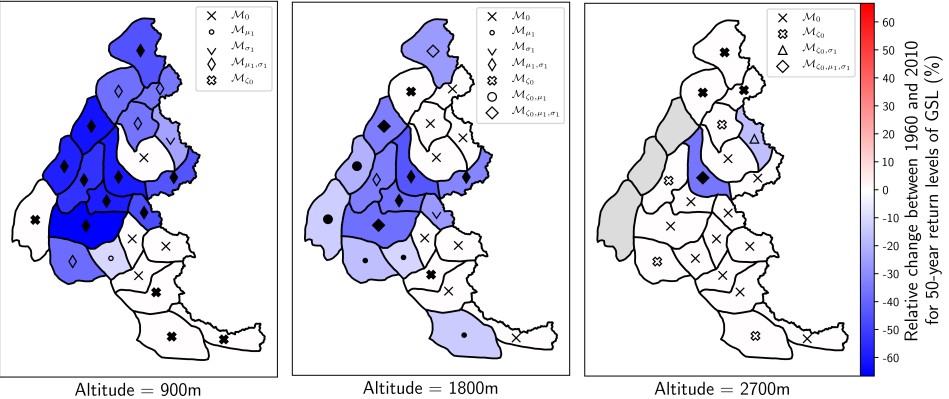

**Figure 5.** Trends in 50-year return levels of ground snow load (GSL) between 1960 and 2010 at low (900 m), mid (1800 m) or high (2700 m) altitude. Markers show selected model $\mathcal{M}_N$ while filled markers symbolize models that are significantly better than the Gumbel model $\mathcal{M}_0$ (Sect. 4.2). Grey areas denote missing data, e.g. when the altitude considered is above the top altitude of the massif.

Figure 6 emphasizes the evolution of decreasing trends between 900 m and 4800 m of altitude. We observe that decreasing

trends are significant for more than one-third of the massifs, located in the Northwest of the Alps (App. A), until 2100 m (black

bars). In half a century, return levels have dropped on average by up to 30% at 900 m. Until 3300 m, we observe a decline in

the percentage of massifs with a significant decreasing trend. Above 3300 m, we do not find any significant decreasing trend.

For both the relative decrease and the percentage of massifs with a decreasing trend, we notice a similar declining pattern. We

also notice more decline between 3300 m and 3900 m than at 3000 m, which echoes results from Lüthi et al. (2019), who found

that, in the Alps above 3000 m, the relative decrease for projected winter-mean of fresh SWE is more marked than at 3000

m (see their Figure 8). We emphasize, however, that most meteorological observations used as input to the SAFRAN-Crocus

reanalysis are situated below 2000 m. Therefore, trends beyond 2000 m altitude should be considered with great caution.

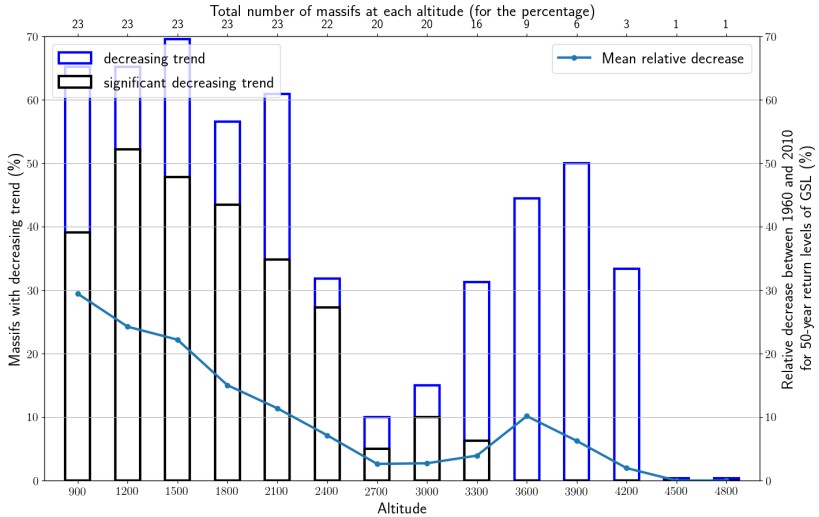

**Figure 6.** Temporal decreasing trend of 50-year return levels of ground snow load (GSL) between 900 m and 4800 m of altitude.

Figure 7 illustrates that, for altitudes 300 m and 600 m, in general no trends are found except few decreasing trends at 600

235 m, and 2 time series (1 at 300 m, 1 at 600 m) with important increasing trends (+100% for one massif at 600 m). Despite

this important increase in relative change, annual maxima of snow load remain small ($< 1 \, \mathrm{kN} \, \mathrm{m}^{-2}$). Indeed, we found that

these annual maxima correspond to snow load accumulated in few days, and thus are mainly driven by heavy precipitation

rather than a seasonal snowpack accumulation. In particular, we hypothesize that the important increasing trend observed in

the South at 600 m (color red) might be cause by a regional phenomenon, resulting from Mediterranean humid air masses

flowing northward into the North of Italy and then westward to the eastern part of the French Alps, that might be intensifying

with global warming (Garavaglia et al., 2010; Gottardi et al., 2012; Faranda, 2019).

To sum up trends in return levels of ground snow load: from 900 m to 4800 m, either no trends or decreasing trends of

50-year return levels of GSL are found (Fig. 5, Fig. 6, Fig. A1), while at 300 m and 600 m, no clear trends are found (Fig. 7).

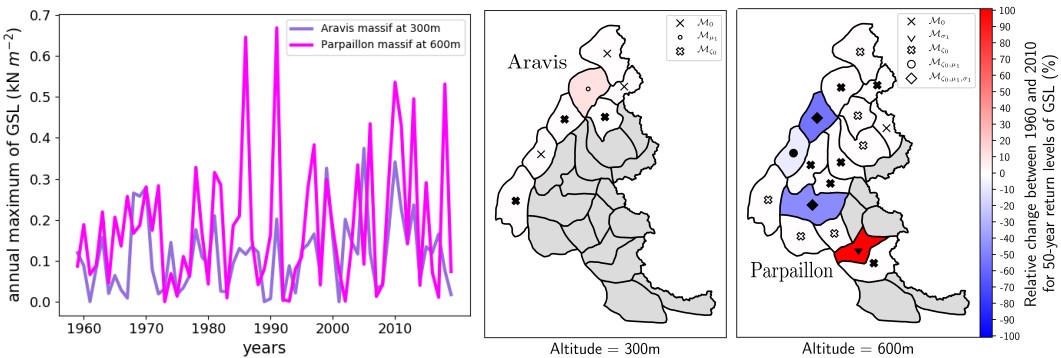

**Figure 7.** Left: Time series of annual maxima for ground snow load (GSL) for 2 massifs (Aravis and Parpaillon) either at 300 m or 600 m of altitude. Right: Trend for annual maxima of ground snow load (GSL) at 300 m and 600 m of altitude. Markers show selected model $\mathcal{M}_N$ while filled markers symbolize models that are significantly better than the Gumbel model $\mathcal{M}_0$ (Sect. 4.2). Grey areas denote either time series that were excluded (Sect. 4.4) or missing data, e.g. when the altitude considered is above the top altitude of the massif.

## 5.3 Comparison of return levels of ground snow load with French standards

We compare 50-year return levels of GSL and their uncertainty (Sect. 4.3) to French standards first for 2 massifs (Fig. 8) then globally (Fig. 9). We consider GSL data from 300 m to 1800 m because standards are defined from 200 m to 2000 m (Sect. 3).

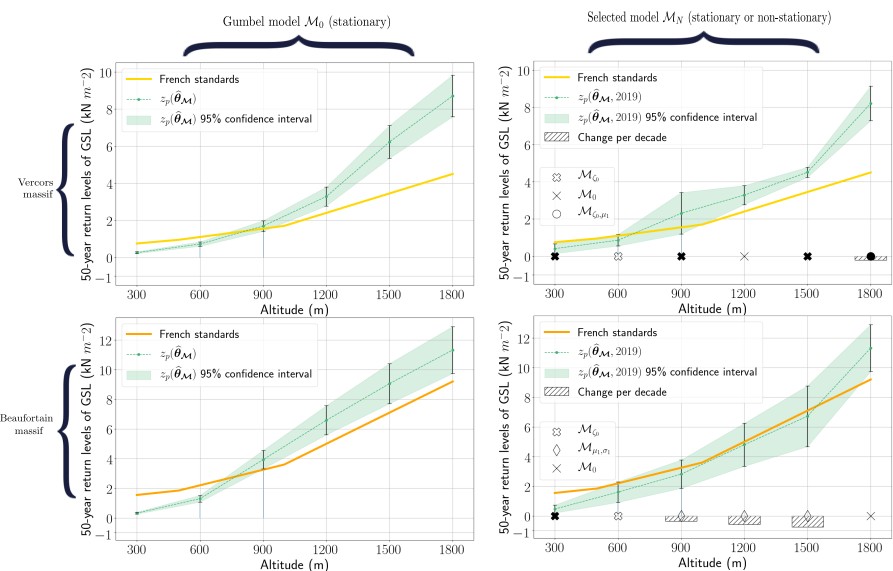

**Figure 8.** 50-year return levels of ground snow load (GSL) from altitude 300 m to 1800 m for Vercors (top) and Beaufortain (bottom) massifs. Return levels (green line) and their uncertainty (shaded green and black bars) are estimated from the data either with the stationary Gumbel model $\mathcal{M}_0$ (left) or with the selected model $\mathcal{M}_N$ (right). If $\mathcal{M}_N$ is a non-stationary model, the return level is the effective return level in 2019, and we display the change of return levels per decade (striped histogram), i.e. 10 times the time derivative of return level (Sect. 4.3).

Figure 8 illustrates these levels and their uncertainty for two massifs (Vercors and Beaufortain) associated with different French standards regions. Standards are often exceeded at higher altitudes (e.g. at 1800 m). Also, Figure 8 exemplifies the impact of accounting for decreasing trends in return levels. Indeed, we observe that return levels from the stationary Gumbel model $\mathcal{M}_0$ (left) are often larger than effective return levels in 2019 (last year of data) from the selected model $\mathcal{M}_N$ (right).

Figure 9 sums up the comparison between French standards and 50-year return levels for all 23 massifs. We display (i) the percentage of massifs whose return level exceeds standards, and (ii) the mean relative difference between return levels and standards. Limits of the confidence intervals are approximated as the percentage of exceedances (resp. mean relative difference) for the limits of return levels' 95% confidence interval (black bars in Fig. 8) and displayed with black bars (resp. shaded blue). The number of massifs considered is equal to 7 at 300 m, 17 at 600 m, and 23 at 900 m and above.

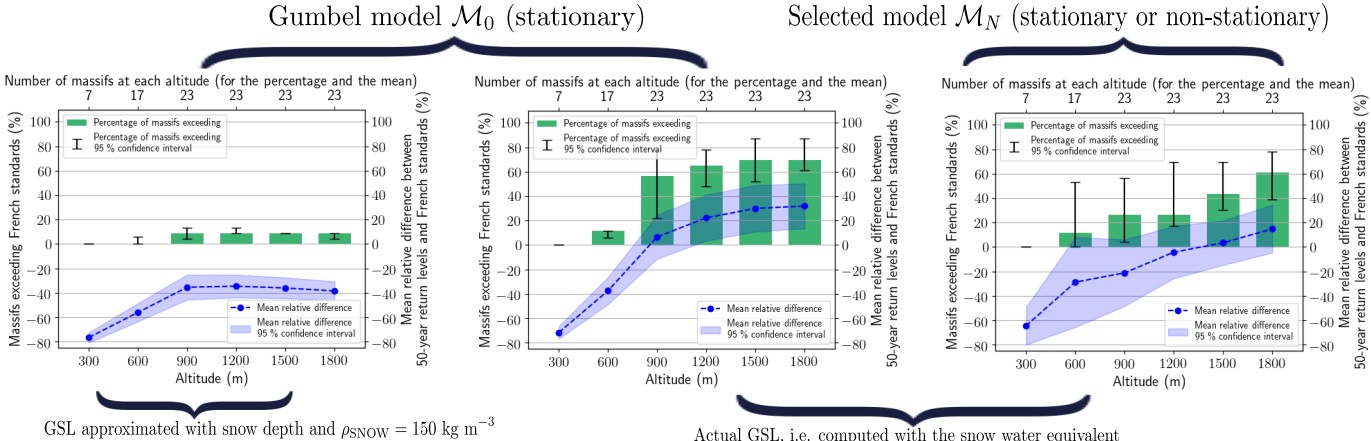

**Figure 9.** Comparison of 50-year return levels of ground snow load (GSL) with French standards between 300 m and 1800 m. We show the percentage of massifs (green histogram) whose return levels exceed French standards, and the mean relative difference (blue line) between return levels and standards. Left: similar to French standard estimation (stationary Gumbel $\mathcal{M}_0$, and GSL approximated with snow depth obtained from the reanalysis and $\rho_{SNOW} = 150$ kg m$^{-3}$). Center: stationary Gumbel $\mathcal{M}_0$, and actual GSL, i.e. computed with SWE from the reanalysis. Right: selected model $\mathcal{M}_N$ (if $\mathcal{M}_N$ is non-stationary, the return level is the effective return level in 2019) and actual GSL.

First, if we estimate return levels from data with the French standards method (Fig. 9 left), i.e. with a stationary Gumbel model $\mathcal{M}_0$, and GSL data approximated with snow depth obtained from reanalysis and $\rho_{SNOW} = 150$ kg m$^{-3}$, then we observe few exceedances (less than 10%) and that on average return levels remains below standards, as the mean relative difference remains below zero. Thus, in this setting, estimations from our reanalysis are consistent with French standards.

However, if we consider the actual GSL, i.e. computed with SWE from the reanalysis, then French standards drastically underestimate return levels. Indeed, with a stationary Gumbel model $\mathcal{M}_0$, then for altitudes above or equal to 900 m, French standards are exceeded for a majority of massifs (Fig. 9 center). But, if we consider the selected model $\mathcal{M}_N$, i.e. if we account for the decreasing trend in 50-year return levels, we have less exceedances at all altitudes (Fig. 9 right). In the latter case, at worst, i.e. at 1800 m, return levels exceed standards by 15% on average, and half of the massifs (60%) exceeds standards.

Furthermore, despite that uncertainty intervals (black bars) can be large, it does not impact the main conclusions of this article. Indeed, in Figure 9 right at 1800 m, we still have between 40% and 80% of massifs exceeding French standards.

## 6    Discussion

### 6.1    Methodological considerations

We discuss in depth the statistical models chosen for this study. It is well-known that an annual maximum based approach
can be wasteful in terms of data (Coles, 2001). However, since our objective is to estimate 50-year return levels and since we have 60 years of data, we still decide to rely on the annual maximum based approach (with the GEV distribution) rather than on the concurrent approach based on threshold exceedances (with the Generalized Pareto distribution). Also, with the GEV distribution, our methodology is a direct extension of French building standards (Sect. 3).

      For the non-stationary models, we focus on simple deterministic functions of time $(\mu(t), \sigma(t), \zeta(t))$ due to the limited length
of time series. A linear non-stationarity seems preferable to a non-stationarity based on the Heaviside step function due to the continuous nature of climate change. We start the linear non-stationary at the initial year, i.e. 1959.

      We decided to consider non-stationarity only for the location and scale parameter. Indeed, in the literature, a linear non-stationarity is considered sometimes only for the location parameter (Fowler et al., 2010; Tramblay and Somot, 2018) but more often both for the location and the scale (or log-transformed scale for numerical reasons) parameters (Katz et al., 2002; Kharin
and Zwiers, 2004; Marty and Blanchet, 2012; Wilcox et al., 2018). Also, we consider a non-stationarity for both parameters because the scale parameters were not proportional to the location parameters, which could have otherwise simplified our parametrization. Finally, the shape parameter is typically considered constant in the literature, and we follow this approach.

      For time series containing zeros, French standards rely on a mixed discrete-continuous distribution. They fit both a Gumbel distribution on non-zero annual maxima and the probability of having a non-zero annual maxima. However, with our reanalysis
data, this approach sometimes leads to fitting non-stationary extreme value models with less than 40 non-zero annual maxima. Therefore, we rather decide to exclude any time series with more than 10% of zeros (Sect. 4.4), to ensure that we fit models with more than 55 non-zero annual maxima. In practice, our approach gives 50-year return levels close to the approach from French standards (absolute difference remains lower than $0.1$ kN m$^{-2}$).

### 6.2    On the limitation to approximate annual maxima of ground snow load with annual maxima of snow depth

SWE times the gravitational constant equals GSL. However, most countries do not measure SWE but only have access to snow depth (HS) (Haberkorn et al., 2019). In that case, snow density is required to obtain SWE (and subsequently GSL) from HS (Sect. 1). In particular, French standards approximate annual maxima of GSL with annual maxima of HS and by assuming a constant snow density, equal to $\rho_{\mathrm{SNOW}} = 150$ kg m$^{-3}$. In Figure 10, we highlight limitations of such approaches with our reanalysis that provides, for the whole snowpack, daily values of SWE, HS, and thus of snow density.

We find that annual maxima of GSL are always underestimated by French standards' approximation (Fig. 10 left). The main reason is that, when annual maxima of GSL are reached, snow density is on average largely superior to $\rho_{\text{SNOW}} = 150 \text{ kg m}^{-3}$ (Fig. 10 center). Indeed, we observe that at the time of the annual maxima of GSL the snow density is around $\approx 350 \text{ kg m}^{-3}$ on average at 2700 m, and close to $\approx 250 \text{ kg m}^{-3}$ on average at 900 m. Finally, despite high variations along the years, we also notice that, when annual maxima of GSL are reached, snow depth can be much lower than the annual maxima of snow depth (Fig. 10 right), which is another argument against the use of snow depth maxima as a proxy for GSL maxima.

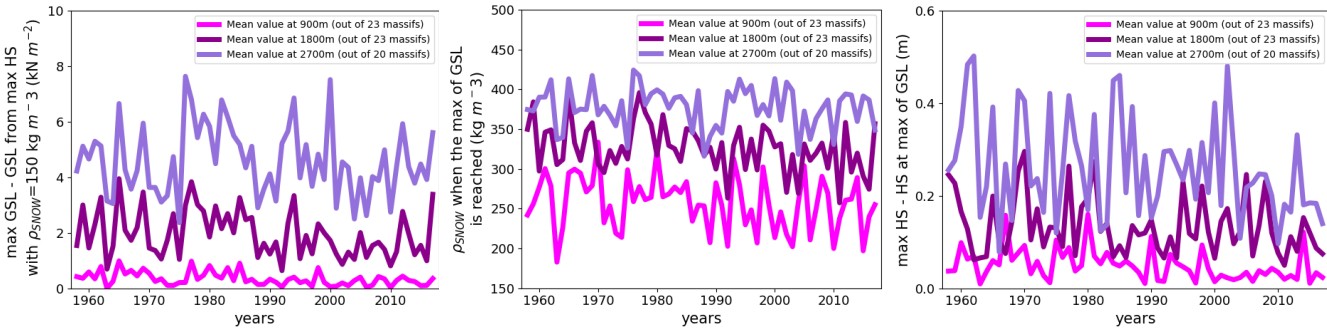

**Figure 10.** Limitation of approximating annual maxima of ground snow load (GSL) from annual maxima of snow depth (HS). Left: Difference between annual maxima of GSL and GSL computed from annual maxima of HS and $\rho_{\text{SNOW}} = 150 \text{ kg m}^{-3}$. Center: Snow density when annual maxima of GSL are reached. Right: Difference between annual maxima of HS and HS when annual maxima of GSL are reached.

## 7  Conclusions

Based on both a reanalysis and a snowpack model, we detect an overall temporal decreasing trend of 50-year return levels of ground snow load (GSL) between 900 m and 4200 m, which is significant until 2100 m in the Northwest of the French Alps. This confirms other studies in the Western Alps which also found overall decreasing trends in linked snowpack variables: SWE and snow depth. The largest decrease is found at 900 m with $-30\%$ for return levels between 1960 and 2010. Despite these decreases, in 2019 return levels still exceed return levels designed for French building standards under a stationary assumption. At worst, i.e. at 1800 m, return levels exceed standards by $15\%$ on average, and half of the massifs exceeds standards.

We hypothesize that this amount of exceedances might be due to an underestimation of GSL by French standards. Indeed, these standards were devised with GSL estimated from snow depth maxima and constant snow density equal to $150 \text{ kg m}^{-3}$, which underestimate typical GSL values for the snowpack. Another reason for these exceedances might be ill-designed relationships between altitude and snow load. As shown in Fig. 2, French standards return levels increase linearly with altitude in three steps. Indeed, French standards (Biétry, 2005) follow previous national standards (AFNOR, 1996) that advised for a linear relationship between altitude and snow load instead of relying on European standards' results that showed a quadratic relationship for the Alpine Region (Sanpaolesi et al., 1998). Thus, at higher altitudes, French standards underestimate actual return levels which might explain the augmenting percentage of exceedance observed with the altitude (Fig. 9 right).

Many potential extensions of this work could be considered. First, our methodology could be extended with more advanced definitions of non-stationary return levels (Rootzén and Katz, 2013; Serinaldi, 2015). Also, instead of considering time series of annual maxima as spatially independent, we believe that our analysis may benefit from an explicit modelling of the spatial dependence between extremes. Then, reanalyses are increasingly available at the European scale (e.g. Soci et al. 2016), which could be used for extending this work to a wider geographical scale. This requires, however, to remain cognizant of the limitations of such reanalyses, in particular (i) the temporal heterogeneity of the meteorological data input to these reanalyses (Vidal et al., 2010), (ii) the lack of observations at high altitudes, requiring caution in analyzing trends for high altitude locations and (iii) model errors (e.g. snowpack model errors) which need to be taken into account when analyzing the results.

Finally, even if, according to our analysis, GSL exceeds French standards return levels in the French Alps, (Fig. 9 right), few destructions related to snow loads actually occurred. Several reasons might explain that. First, French standards consider a coefficient that maps GSL return levels to roof snow load return level, i.e. multiplication by a coefficient that summarizes several roof features: shape, exposure and thermal transmission (Sanpaolesi et al., 1998). This coefficient might be overprotective. Also, following European standards, roof designers must add safety coefficients to ensure roofs' reliability. Indeed, they actually build roofs that withstand the sum of (i) the characteristic value of permanent action, i.e. self-weight, multiplied by a safety coefficient equal to $1.35$ and (ii) the characteristic value of variable action, i.e. roof snow load return level, multiplied by a safety coefficient equal to $1.5$ (Sanpaolesi et al. 1998 Eq. 8). Above all, French standards do not take into account that, after intense days of snowfall, the snow accumulated on the roof either slides off or is removed. In that case, the main risk lies in extreme snow events that might accumulate in few days enough snow to exceed French standards. Undeniably, most known snow load destructions resulted from such intense snow events, sometimes combined with liquid precipitation that often heavily increase snow load. The response of these short but extreme and complex snow events to climate change might be an interesting topic for future research.

*Author contributions.* E. L. R. performed the analysis and drafted the first version of the manuscript. All authors discussed the results and edited the manuscript.

*Competing interests.* The authors declare that they have no conflict of interest.

*Data availability.* The dataset can be download from AERIS website https://www.aeris-data.fr/catalogue/ (type "S2M" in the search bar).

*Acknowledgements.* E. L. R. holds a PhD grant from INRAE. The S2M data are provided by Météo-France - CNRS, CNRM Centre d'Etudes de la Neige, through AERIS. We are grateful to Eric Gilleland for his "extRemes" R package. Finally, we are indebted to Jacques Biétry for providing us the report on French standards w.r.t. ground snow load and for his explanations on their methodology.

## Appendix A: Trends in return levels of ground snow load

In this section, we report, for every 300 m of altitude from 900 m to 4200 m, the map of the relative change of 50-year return levels of GSL between 1960 and 2010 (Fig. A1). Trends at 4500 m and 4800 m are not reported, since they only concern the Mont Blanc massif, where no significant trend is inferred at these altitudes.

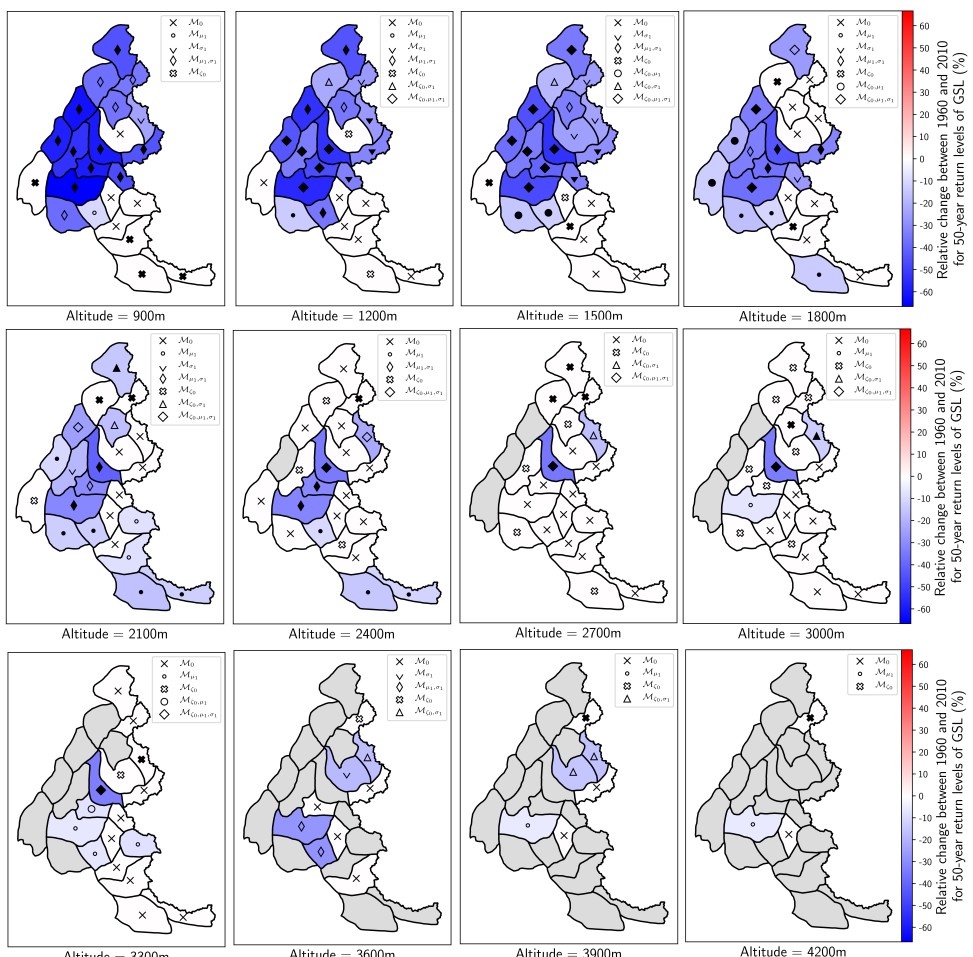

**Figure A1.** Trends in return levels of ground snow load (GSL) between 900 m and 4200 m of altitude. Markers show selected model $\mathcal{M}_N$ while filled markers symbolize models that are significantly better than the Gumbel model $\mathcal{M}_0$ (Sect. 4.2). Grey areas denote either time series that were excluded (Sect. 4.4) or missing data, e.g. when the altitude considered is above the top altitude of the massif.

### Appendix B: Detailed methodology for the model validation

**Quantile-quantile plot**. Standard diagnosis tools for both stationary and non-stationary extreme value models (Coles, 2001; Richard W. Katz, 2012) rely on a probability integral transformation $f$ to the standard Gumbel distribution, i.e. Gumbel$(0, 1)$. Indeed, if $Z(t) \sim \text{GEV}(\mu(t), \sigma(t), \zeta(t))$, then $f(Z(t)) = \frac{1}{\zeta(t)} \log(1 + \zeta(t) \frac{Z(t) - \mu(t)}{\sigma(t)}) \sim \text{Gumbel}(0, 1)$. Thus, if $\boldsymbol{z} = (z_{1959}, ..., z_{2019})$ represent a time series of annual maxima, then let $\tilde{z}_{1959} = f(z_{1959}), ..., \tilde{z}_{2019} = f(z_{2019})$.

Quantile-quantile (Q-Q) plot is a standard diagnosis based on the comparison of empirical quantiles (computed from the empirical distribution) and theoretical quantiles (computed from the expected distribution). On one hand, $\tilde{z}_{(1)}, ... \tilde{z}_{(61)}$ are the empirical quantiles, which correspond to the ordered values of the $\tilde{z}_t$. On the other hand, $-\log(-\log(\frac{1}{62})), ..., -\log(-\log(\frac{61}{62}))$ correspond to the theoretical quantiles. Indeed, if $\tilde{Z} \sim \text{Gumbel}(0, 1)$, then $P(\tilde{Z} \leq \tilde{z}) = \exp{-e^{-\tilde{z}}} = \frac{i}{62} \leftrightarrow \tilde{z} = -\log(-\log(\frac{i}{62}))$. Thus, the Q-Q plot is comprised of the pairs $\{(-\log(-\log(\frac{i}{62})), \tilde{z}_{(i)}); i = 1, ..., 61\}$.

In Figure B1, we display Q-Q plots for the three time series of annual maxima of GSL displayed in Fig. 1. We observe that the left and the right Q-Q plots show a good fit, as the points stay close to the line. However, for the center Q-Q plot, all points are close to the line, except the highest empirical quantile that is largely above the corresponding theoretical quantile. As a whole, when observing all Q-Q plots (not shown) most time series show a good fit, except few time series (less than 10) which have a pattern similar to the center Q-Q plot in Figure B1.

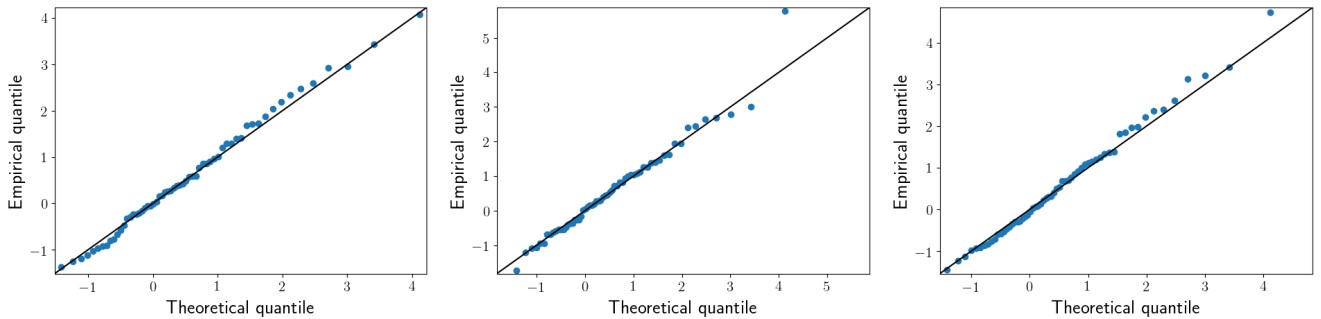

**Figure B1.** Q-Q plots of the selected models for the three time series displayed in Fig. 1. Left: Ubaye massif at 900m fitted with the model $\mathcal{M}_{\zeta_0}$. Center: Vercors massif at 1800m fitted with the model $\mathcal{M}_{\zeta_0, \mu_1}$. Right: Beaufortain massif at 2700m fitted with the model $\mathcal{M}_{\zeta_0}$.

**Anderson-Darling test**. Q-Q plot is a qualitative tool to validate the goodness-of-fit for probability models. For the quantitative validation of the goodness-of-fit of the selected models, we rely on the Anderson-Darling statistical test, which is the most powerful test for the Gumbel distribution according to the comparative study of Abidin et al. (2012).

In practice, with this test, we assess if the transformed annual maxima $\tilde{z}_{(1)}, ..., \tilde{z}_{(61)}$ are likely to be generated from a standard Gumbel distribution. Let $n = 61$ denotes the number of samples, and $F_{\text{emp}}$ (resp. $F_{\text{gum}}$) denotes the cumulative distribution function of the empirical (resp. standard Gumbel) distribution. Then, Anderson-Darling test is based on the distance:

$$A^2 = n \int (F_{\text{emp}}(x) - F_{\text{gum}}(x))^2 w(x) dF_{\text{gum}}(x) \approx -\sum_{i=1}^{n} \frac{2i-1}{n} \{\log[F_{\text{gum}}(\tilde{z}_{(i)})] + \log[1 - F_{\text{gum}}(\tilde{z}_{(n+1-i)})]\} - n. \tag{B1}$$

where $w(x)$ places more weight on the tail of the standard Gumbel distribution. For details, we refer to Abidin et al. (2012).

We apply this test on the transformed data using Saeb (2018), and found that we cannot reject the null hypothesis (samples generated from the Gumbel model) at the 5% significance level for almost all our selected models (98%), justifying their good fit. As explained in Sect. 4.4, we exclude time series whose selected models do not pass this Anderson-Darling test.

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
