# Peer review of "Non-stationary Extreme Value Analysis of Ground Snow Loads in the French Alps: a Comparison with Building Standards"

_Natural Hazards and Earth System Sciences, 2020_

## Referee Comment (RC1) · Anonymous Referee #1 · 25 Apr 2020

The paper presents a comparison of models for annual maxima of the ground snow loads (GSL) for each massif and each altitude interval of 300 m. The simplest models are the Gumbel and GEV with fixed parameters, then come non-stationary GEV models with either the location or location and scale parameters that may vary linearly with the year. The GLS data is provided by feeding the snow pack model Crocus with Safran meteorological reanalyses. According to the authors, only analyses with a proxy for GLS (such as snow depth) were carried out previously. The statistical models for extreme values applied in this work are relatively simplistic. This is justified by the authors by the short duration of the reanalysis data. The quantity of interest is the 50 year return level as it is used by the French regulation. A thourough comparison of the

estimates provided by the analyses in the paper with the French standard is conducted.

I missed some validation or references to validation of the GLS data. As mentioned by the authors, Safran has a number of biases. Crocus might be based on assumptions which are not alway fulfilled and so the end product, GLS, might also suffer from a number of shortcomings.

In addition, there is no validation of the GEV models, just the final selection among the models in Table 2. These are based on AIC and likelihood ratios. So the best model is selected. But do they fit well ? What if none of the models were really adequate (even the best one among them) ? Maybe some qq-plots analyses should be included.

Given the amount of literature, I found it a bit disappointing that no attempt was made to rely on models that make use of more data, not only annual maxima as mentioned in the discussion. For instance, the tail index is taken to be constant in view of the difficulty to estimate it. There are many ways around this, one of which is the so-called regional analysis.

The authors argue that the number of years of the GSL reanalysis is too short to attempt to use anything else than linear relationships in the non-stationary models. Nevertheless, they recognize that other extreme value approaches, such as peaks-over-threshold, can be apply to exploit more data (more than a maxima per year). This seems a bit contradictory. If the authors could show that the GEV models with linear non-stationarities fit well the data without too much uncertainty in the estimates, then it would alleviate this issue.

Although the paper is generally well written, I think it can be improved on a number of aspects. - I found that the abstract was not conveying too well the main analyses and conclusions of the paper. - Section 2 : Line 57 : "we rely on GSL reanalysis", please, specify that is is a reanlysis of GSL. - P.3 What is the spatial resolution of Safran ? Does GSL has the same spatial resolution ? -P.7 L.134 : somewhere in the likelihood ratio, the model $M_0$ should appear. Check the expression, I think there is a mistake.

P.11 last sentence : "... often above effective return levels " effective in what ways ? not sure what it means here -P.13 L.245-250 : " ... start the non-stationarity after the most likely year ", what is meant by most likely year ? -P.14 L. 260 "We did not rely on this choice because ..." This sentence should be rephrase, it is not clear, in my opinion. -P.14 L.265-270 : "annual maxima of GSL ARE " ARE instead of is, in several places - P.14 L.265-270 : "THE main reason is that " use THE - P.14 L.265-270 but in other places as well : ON average not in average - P.15 the trends OF the 50-year return levels not "w.r.t"

---

## Referee Comment (RC2) · Anonymous Referee #2 · 25 May 2020

Dear authors

I am happy to see this field getting attention. The paper is nice to read and due to the simple language mostly easy to follow. It provides valuable results certainly worth to be published. However, prior to final publication I recommend consideration of one major point and a number of technical corrections summarized below.

General comments:

My main concern is about a missing (short) elaboration on the data the study is based on. I miss a validation of the GSL values (or at least a description of the errors) produced by the reanalysis of Vernay (2019), in particular that of the yearly maximum

values. In addition it would be nice to have an explanation of how GSL values are assigned to the massif scale. Moreover, the text contains many spelling and grammar mistakes. However most of them are repetitive.

Specific comments:

The manuscript lacks a description of (i) error measures of GSL data used as basis of the extreme value statistics, and (ii) general remarks on the reanalysis used to provide that data. In particular it would be crucial to tell something about the BIAS or absolute errors of the yearly maximum GSL values. Otherwise provided uncertainty assessments are less valuable. Furthermore a general description of some aspects of the reanalysis is missing. How is GSL calculated for the massif scale? Is the 50-year GSL return level computed by your models valid for the whole massif just depending on altitude? The abstract of Vernay (2019) states also a dependency on aspect and slope. You should clarify if your results are valid for distinct elevations or elevation bands (as it is stated here and there). In the latter case you should explain, how GSL values are assigned to that band (see lines 59, 71 in your manuscript).

Technical corrections:

I added a list of technical corrections. Most of them are language related.

A general comment about the figures: Smaller fonts in plots are difficult to read even with 200% enlargement (e.g. Fig. 8).

8: . . .from snow depth _maxima_ and constant. . .

9: You do not talk about layers in the snowpack, so it is clear that always the full snowpack is concerned. I suggest to remove _full_ (also in line 281)

12: I would the year 2006 not call recently...

20: This is where snow water equivalent is introduced. The short form SWE could be added here instead of several other places in the manuscript (caption of Table 1, lines

64, 262).

25: past trends . . . _show_ a decreasing trend.

29: check format of first reference

32: Table 1: Column name is "Trend", in the caption it is "trend".

36: remove i.e.; _on_ instead of in average

48: _of 50-year return levels_ instead of _in 50-year return_

60: _denote_

70: remove _the_ before SWE

75: Caption of Figure 2: w.r.t. altitude. Remove _the_

76: _at_ stations, not in

78: _the_ characteristic

80: As maximum values are relevant in this study, the procedure of _removing the top annual maximum when considered exceptional_ should be shortly addressed. I can imagine that one can find information about that in the given reference, but this is in French. . .

84: What exactly do you mean with _relative change_? Relative to what? (see also line 48)

85: _of_ not in

89: replace rarer extremes with _extremes that are more rare_ or similar; parenthesis: (EVT, Coles, 2001)

112: remove _that_; correspond_s_ to

119: represent, not represents; its not an elevation _band_ but a distinct altitude, right?

(see also lines 59, 71, 123)

122, 126: I wonder if these complex expressions are necessary to understand the content? If not you could remove them.

130: this test _assesses_; you could replace the sloppy term _move_ with something more statistical like _to reject the stationary Gumbel model M0 in favor of the selected model MN _ or similar.

134: Shouldn't there somewhere be the M0 model in the expression of the LR test?

136: remove _it means that_

137: If the selected…; remove _then_

138: Return level_s_

141: _on_ average

150: For the selected model MN the return level is defined as…

151: remove _that_

159: associated_with_ the maximum

161: Therefore, under regularity: remove _we have that_

168-169: fit models _to_ time series

169: only those _models_; we _make_ this choice

173: Result_s_

177: Gumbel models are always more _often_ selected _than_ their

180: model_s_; that _the_ majority

181: _are white colored_: white is not a color. At the end of the sentence: replace _It_ with _This_

184-185: You should rephrase the sentence "This uncertainty. . ..robustly."

186: _In_ instead of On figure

187: The altitude in Fig. 4 is 1800 m, not 1500 m

188-191: Your hypothesis was already raised for snow depth by Blanchet & Lehning, 2010, Blanchet & Davison, 2011 and Schellander et al., 2018, who all found similar results in terms of the shape parameter.

Figure 4: The legend is somewhat hard to read. I guess these are jpgs or similar, which should be replaced by pdfs. The grey color in the rightmost panel has no description. In the caption: model_s_; significant _trends_.

196-197: you say: "Gumbel or GEV distribution (diamond-shaped filled markers)." But diamond shaped filled markers describe only Gumbel models, and not GEV models.

197: you say "no trends", but actually I see one trend (green colored massif in the south at 2700m); _white color_: white is no color

Figure 5: Grey color has no description (see Figure 4). In the caption: symbolize _a_significant trend.

200: Sect. A should be Appendix A

201: _on_average

201-202: What is a null decline? I think a decline is always non-null, isn't it?

204: The word "growth" is misleading (growing decline?). You could instead say "less declining"; Lüthi et al. (2019) _who_ found

208: trend_s_

209: What do you mean with "sometimes important" here?

210: You mention "recorded annual maxima". Do you compare with observed GSL

values? How are they recorded? If manually (weekly or biweekly) you cannot be sure to get the maximum. Or does recorded mean "modeled" here?

212: precipitation not precipitation_s_; season_al_

212-213: "the 2 massifs with important increasing trends (red color) might be caused". The trends may be caused but not the massifs.

216: decreasing trend_s_; comma instead of dot after parenthesis in "(Fig. 5, Fig. 6 and Fig. A1). While"

Figure 7: No description for grey color.

219: Why of all things 1800 m? Is this because Vercors top heights are around 1800 m?

221: associated _with_

224: (_l_eft); replace above with _larger than_; (_r_ight)

225: While reading and coming from Fig. 8 one assumes that Fig. 9 sums up only two massifs. You could state here that Fig. 9 encompasses all 23 massifs.

228: There should only be one dot at the end of the sentence.

229: _the_ French standards; _l_eft

232: remove "the" in "…computed with with the snow water equivalent…."; _the_ re-analysis

234: _c_enter

235: _r_ight

236: _more than_ half of the massifs

Figure 8: I suggest to put parts of the legend in the caption for a cleaner layout. E.g. all additional information with respect to the left plot except the "Change per decade". Top

left panel: Do you have a clue, why the uncertainties at lower altitudes are larger than at higher altitudes? With respect to the smaller number of available reanalysis stations at higher altitudes, this should be inverted, as can be seen in all other panels. Caption: All panel references should start with a small letter (_top_ instead of Top . . .). Several times _the_ return level. The change in return level_s_ per decade

Figure 9: Title: "Stationary Gumbel model M0 (stationary)": I suggest to omit Stationary at the beginnig. Caption: _larger than_ instead of above. Remove _the_ in ". . .computed with the snow water equivalent. . .". It should be _the_ reanalysis (2 times) and _the_ return level.

245: we focus on _a_ simple function

257-258: You obtained the "same" results for time series with less than 10% of zero GSL values. Can you provide a similar number used by French standards for the decision to switch to a mixed discrete-continuous distribution?

267-273: This paragraph is a little bit confusing. Descriptions in the text do not match the plots parameters and captions. The easiest way to fix that would be to interchange left and right panels.

267: HS instead of GSL? Annual maxima _are_; _The_ main reason; _l_eft etc.

268: _are_reached; _on_ average

269: _c_enter

270: _on_ average (2 times)

271: _are_ reached

272: _r_ight

280: estimated from _maximum_ snow depth_s_?

281: What do you mean with "full snowpack"? You could use only "snowpack" or "bulk

density", which refers to the density of the whole snowpack.

282: _in_ Fig. 2; You could say e.g. "French standards return levels increase linearly with altitude in three steps" instead of "French standards return levels augment linearly by parts w.r.t the altitude."

285: Remove "might" in "French standards might underestimate"

286: percentage of exceedance _observed with_ altitude; _r_ight

288-289: considering time series of annual maxima as _spatially_ independent

290: reference without parenthesis: (e.g. Soci et al., 2016)

291: _to_ a wider geographical scale

294: remove comma in (e.g., snowpack model errors)

295: _r_ight297: return level_s_

298: remove second parenthesis

299-300: This statement is unclear. I suggest to either remove it, or to provide more details. If you really would like to leave that here, you should provide at least a reference for the European construction standards, and elaborate a little bit on those safety coefficients that might alter very widely according to country, professional, construction material, etc.

306, 310: E.L.R or ELR

309: _The_ dataset

311: for _his_ "extRemes" package

Figure A1: No explanation of the grey massifs.

2020-81, 2020.

---

## Author Comment (AC1) · 25 May 2020

Response to Interactive comment by Anonymous Referee #1

We thank the referee for this thorough review and for the numerous constructive suggestions. Please find below, our answers to these suggestions.

**2. I missed some validation or references to validation of the GLS data. As mentioned by the authors, Safran has a number of biases. Crocus might be based on assumptions which are not always fulfilled and so the end product, GLS, might also suffer from a number of shortcomings.**
*We will add several references emphasizing that SAFRAN and Crocus have been well validated.*

**3. In addition, there is no validation of the GEV models, just the final selection among the models in Table 2. These are based on AIC and likelihood ratios. So the best model is selected. But do they fit well ? What if none of the models were really adequate (even the best one among them) ? Maybe some qq-plots analyses should be included.**
*Quantile-Quantile (Q-Q) analysis is performed for all selected models. To apply this analysis to both stationary and non-stationary model, we rely on [1] that suggests 1) to transform the data to stationary Gumbel 2) to use a Q-Q plot analysis on the transformed data w.r.t. to a Gumbel distribution. Q-Q plots reveal that transformed data is well fitted by a stationary Gumbel distribution, hence that data is well fitted by the selected models.*
*Moreover, according to the comparative study [2], the most powerful Goodness of Fit test for the Gumbel distribution is a combination of the Anderson-Darling test and the Maximum Likelihood Estimator. We apply this test on the transformed data, and found using [3] that we cannot reject the null hypothesis (samples generated from the Gumbel model) at the 5% significance level for 98% of the time series, justifying the good fit of our selected models.*
*We will add these test results at the beginning of the Result section. In an Appendix section, we will detail an explanation on the methodology of [1], and display Q-Q plots for the time series presented in section 2.*

**4. Given the amount of literature, I found it a bit disappointing that no attempt was made to rely on models that make use of more data, not only annual maxima as mentioned in the discussion. For instance, the tail index is taken to be constant in view of the difficulty to estimate it. There are many ways around this, one of which is the so-called regional analysis.**
*SAFRAN reanalysis are the result of a postprocessing of the meteorological observations at the massif scale and, as such, already represents "regionalized" data. In this context, it does not seem clear how a regional analysis could be performed.*

**5 The authors argue that the number of years of the GSL reanalysis is too short to attempt to use anything else than linear relationships in the non-stationary models. Nevertheless, they recognize that other extreme value approaches, such as peaks-overthreshold, can be apply to exploit more data (more than a maxima per year). This seems a bit contradictory. If the authors could show that the GEV models with**

linear non-stationarities fit well the data without too much uncertainty in the estimates, then it would alleviate this issue

*Our goal is to implement a clear comparison with French standards. For this reason,,thus we prefer to rely on the Gumbel distribution & extensions of this distribution, which explains our choice to use Gumbel and GEV distributions.*

*Furthermore, the impact of the uncertainty in the estimates is already shown on our main figures (black bars on Figure 9). Despite that these uncertainty interval can sometimes be large, it does impact the main conclusions of this article. For instance, we would still have between 40 and 80% of massifs whose return levels in 2019 exceed French standards.*

6. Although the paper is generally well written, I think it can be improved on a number of aspects

*We will correct expression/syntax mistakes that are mentioned. Also in the modified manuscript, we will clarify the following points:*

6.1. I found that the abstract was not conveying too well the main analyses and conclusions of the paper.

*We will work on improving the abstract once the content of the modified manuscript is finalized.*

6.2. P.3 What is the spatial resolution of Safran ? Does GSL has the same spatial resolution ?

*As explained on l.59, SAFRAN does not provide gridded data, it gives massif-level data. More precisely, as detailed in [6]: " The principle of SAFRAN is to perform a spatialization of the available weather data in mountain ranges with so-called "massifs" of about 1000 km2 where meteorological conditions are assumed to depend only on altitude." In the Data section, we will add a sentence to make that point clear. Otherwise, Crocus snowpack model takes SAFRAN data as inputs to produce SWE (which we use to compute GSL), therefore yes, GSL data has the same spatial resolution as Safran.*

6.2. P.11 last sentence : "... often above effective return levels " effective in what ways ? not sure what it means here

*While classical stationary return levels do not depend on time, return levels are denoted as effective when they depend on time [4, 5]. To quote [4]: "[Effective design value] has an interpretation similar to that for an ordinary design value (i.e., the quantile corresponding to a specified return period), except that it varies depending on the time of year.".*

6.3. -P.13 L.245-250 : " ... start the non-stationarity after the most likely year ", what is meant by most likely year ?

*For each model with a linearity in some parameters of the distribution we could choose to start the linearity only after some starting year.*

*The most likely starting year is the year that gives the maximum likelihood for this linear model [6]. However, in the end we decided not to use this approach. Therefore we propose to remove this sentence altogether from the discussion Section to avoid confusing the reader with unnecessary details.*

[1] Richard W. Katz. (2012). Statistical methods for nonstationary extremes. In Extremes in a Changing Climate - Detection, Analysis & Uncertainty (pp. 15–38). Springer Science & Business Media.

[2] Abidin, N. Z., Adam, M. B., & Midi, H. (2012). The Goodness-of-fit Test for Gumbel Distribution: A Comparative Study. Matematika, 28(1), 35–48. Retrieved from http://www.matematika.utm.my/index.php/matematika/article/view/313

[3] Ali Saeb (2018). gnFit R package https://www.rdocumentation.org/packages/gnFit

[4] Katz, R. W., Parlange, M. B., & Naveau, P. (2002). Statistics of extremes in hydrology. Advances in Water Resources, 25(8–12), 1287–1304. https://doi.org/10.1016/S0309-1708(02)00056-8

[5] Mondal, A., & Daniel, D. (2019). Return Levels under Nonstationarity: The Need to Update Infrastructure Design Strategies. Journal of Hydrologic Engineering, 24(1), 04018060. https://doi.org/10.1061/(ASCE)HE.1943-5584.0001738

[6] Nousu, J.-P., Lafaysse, M., Vernay, M., Bellier, J., Evin, G., & Joly, B. (2019). Statistical post-processing of ensemble forecasts of the height of new snow. Nonlinear Processes in Geophysics, 1–32. https://doi.org/10.5194/npg-2019-27

[7] Blanchet, J., Molinié, G., & Touati, J. (2016). Spatial analysis of trend in extreme daily rainfall in southern France. Climate Dynamics, 51(3), 799–812. https://doi.org/10.1007/s00382-016-3122-7

---

## Author Comment (AC2) · 12 Jun 2020

**Response to Interactive comment by Anonymous Referee #2**

We thank the referee for this detailed review and for the numerous suggestions. Please find below our answers.

**Specific comments**: The manuscript lacks a description of (i) error measures of GSL data used as basis of the extreme value statistics, and (ii) general remarks on the reanalysis used to provide that data. In particular it would be crucial to tell something about the BIAS or absolute errors of the yearly maximum GSL values. Otherwise provided uncertainty assessments are less valuable. Furthermore a general description of some aspects of the reanalysis is missing. How is GSL calculated for the massif scale? Is the 50-year GSL return level computed by your models valid for the whole massif just depending on altitude? The abstract of Vernay (2019) states also a dependency on aspect and slope. You should clarify if your results are valid for distinct elevations or elevation bands (as it is stated here and there). In the latter case you should explain, how GSL values are assigned to that band (see lines 59, 71 in your manuscript).

*The SAFRAN-Crocus reanalysis has been evaluated against various observation datasets, as reported in previous publications (Lafaysse et al., 2013, Vionnet et al., 2016, Revuelto et al., 2018, Vionnet et al. 2019). In most cases, the evaluation is carried out against in-situ snow depth observations and remote sensing snow cover information. For example, Vionnet et al., (2016) evaluated SAFRAN-Crocus snow depth data against 79 observed snow depth data in the French Alps for the 2010-2014 time period, with mean bias and standard error values of 18 cm and 37 cm, respectively. This corresponds to typical values for snow modelling systems applied in various regions on Earth. Because of lower data availability, evaluations against observed SWE values are less frequent than against snow depth data, although we note that Crocus has been shown to perform extremely well compared to other snow cover models, in terms of SWE, across many observation sites worldwide (Krinner et al., 2018) and SAFRAN-Crocus exhibits satisfying performance in terms of snow depth and SWE in the Pyrenees (Quéno et al., 2016), providing confidence, with respect to other existing datasets, in using this model chain for ground snow load (GSL) values. Further model evaluations, using additional datasets, are required to continue assessing and improving the quality of the model chain.*

*Furthermore, we highlight that we only used SAFRAN-Crocus reanalysis values on flat field, and we did not used simulations on slopes, hence it is not relevant to discuss the impact of slope and aspect on the results of this study.*

**Technical corrections:**
We will correct expression/syntax mistakes that are mentioned. In the modified manuscript, we will clarify several points including:

80: As maximum values are relevant in this study, the procedure of _removing the top annual maximum when considered exceptional_ should be shortly addressed. I can imagine that one can find information about that in the given reference, but this is in French...

The procedure is as follows: «*If the ratio of the largest load value to the characteristic load determined without the inclusion of that value is greater than 1.5 then the largest load value shall be treated as an exceptional value*» (Sanpaolesi et al., 1998). This will be added to the revised version of the manuscript.

84: What exactly do you mean with _relative change_? Relative to what? (see also line 48)

*We meant "relative change of 50-year return levels of GSL between 1960 and 2010". We will clarify it when necessary, and maybe refer to formula 4 (detailed expression).*

126: I wonder if these complex expressions are necessary to understand the content? If not you could remove them.

*We do not believe that the expression of the AIC is particularly complex. Most importantly, we think that this expression is necessary to understand the model selection, since the penalization of the log-likelihood by the number of fitted parameters clearly appears.*

219: Why of all things 1800 m? Is this because Vercors top heights are around 1800m?

*This is because French standards for extreme snow loads are defined from 200 m to 2000 m (Section 2). As we consider available altitudes between 200 m and 2000 m, only results obtained with reanalysis from 300 m to 1800 m are shown. However, the SAFRAN-Crocus reanalysis can provide results at higher elevation for the mountain areas peaking above 1800 m elevation.*

Figure 8. Top left panel: Do you have a clue, why the uncertainties at lower altitudes are larger than at higher altitudes? With respect to the smaller number of available reanalysis stations at higher altitudes, this should be inverted, as can be seen in all other panels.

*The reviewer must refer to the top-right panel (Vercors massif & Selected model) which is certainly different from the other panel.*

*Indeed, uncertainties usually grow larger with the altitude. Looking at similar plots to Figure 8 for all other massifs (not shown), this pattern is always seen for the left panels, i.e. with the stationary Gumbel model. However, for the right panels, i.e. for the selected model, 6 massifs out of 23 (Vercors, Ubaye, Oisans, Mercantour, Maurienne, Haut Var Haut Verdon) do not present this pattern, i.e. have larger uncertainties at lower altitudes. Some of these uncertainties might be due to variance in the estimated parameters. In particular, the shape parameter of the GEV distribution is known to be difficult to estimate. As shown in Figure 4, at 900 m the Vercors massif (most western massif) is colored in brown, meaning than its shape parameter roughly equals 0.3. This might explain the high uncertainty at 900 m in*

*the top-right panel, as small changes around 0.3 can have large effect in the 50-year return level.*

257-258: You obtained the "same" results for time series with less than 10% of zero GSL values. Can you provide a similar number used by French standards for the decision to switch to a mixed discrete-continuous distribution?

*In the French standards, the mixed discrete-continuous distribution was considered for all time series, those with less than 10%, of zero GSL values, as well as those with more than 10%.*

299-300: This statement is unclear. I suggest to either remove it, or to provide more details. If you really would like to leave that here, you should provide at least a refer-ence for the European construction standards, and elaborate a little bit on those safety coefficients that might alter very widely according to country, professional, construction material, etc.

*We agree with the reviewer that this paragraph should be more detailed, and this will be done in the revised manuscript. Concerning European standards (Sanpaolesi et al. 1998, page 32, equation 8), the design value for the structure equals the sum of i) the characteristic value of permanent action, i.e. self-weight, multiplied by a safety coefficient equal to 1.35 and ii) the characteristic value of variable action, i.e. roof snow load, multiplied by a safety coefficient equal to 1.5.*

**References**

Vionnet, V., Six, D., Auger, L., Dumont, M., Lafaysse, M., Quéno, L., Réveillet, M., Dombrowski-Etchevers I., Thibert, E. and Vincent, C.: Sub-kilometer precipitation datasets for snowpack and glacier modeling in alpine terrain, Front. Earth Sci., 7, 182, https://doi.org/10.3389/feart.2019.00182, 2019.

Vionnet V., Dombrowski-Etchevers I., Lafaysse M., Quéno L., Seity Y., and Bazile, E. : Numerical weather forecasts at kilometer scale in the French Alps : evaluation and applications for snowpack modelling, J. Hydrometeor., 17, 2591-2614, doi:10.1175/JHM-D-15-0241.1

Quéno, L., Vionnet, V., Dombrowski-Etchevers, I., Lafaysse, M., Dumont, M., and Karbou, F.: Snowpack modelling in the Pyrenees driven by kilometric-resolution meteorological forecasts, The Cryosphere, 10, 1571-1589, doi:10.5194/tc-10-1571-2016

Revuelto, J., Lecourt, G., Lafaysse, M., Zin, I., Charrois, L., Vionnet, V., Dumont, M., Rabatel, A., Six, D., Condom, T., Morin, S., Viani, A., and Sirguey, P. : Multi-criteria evaluation of snowpack simulations in complex alpine terrain using satellite and in situ observations, Remote Sensing, 10, 1171, doi:10.3390/rs10081171, 2018.

Krinner, G., Derksen, C., Essery, R., Flanner, M., Hagemann, S., Clark, M., Hall, A., Rott, H., Brutel-Vuilmet, C., Kim, H., Ménard, C. B., Mudryk, L., Thackeray, C., Wang, L., Arduini, G., Balsamo, G., Bartlett, P., Boike, J., Boone, A., Chéruy, F., Colin, J., Cuntz, M., Dai, Y., Decharme, B., Derry, J., Ducharne, A., Dutra, E.,

Fang, X., Fierz, C., Ghattas, J., Gusev, Y., Haverd, V., Kontu, A., Lafaysse, M., Law, R., Lawrence, D., Li, W., Marke, T., Marks, D., Nasonova, O., Nitta, T., Niwano, M., Pomeroy, J., Raleigh, M. S., Schaedler, G., Semenov, V., Smirnova, T., Stacke, T., Strasser, U., Svenson, S., Turkov, D., Wang, T., Wever, N., Yuan, H., and Zhou, W.: ESM-SnowMIP: Assessing snow models and quantifying snow-related climate feedbacks, Geosci. Model Dev., 11, 5027-5049, https://doi.org/10.5194/gmd-11-5027-2018, 2018.

Lafaysse, M.., S. Morin, C. Coléou, M. Vernay, D. Serça, F. Besson, J.-M. Willemet, G. Giraud and Y. Durand, 2013 : Towards a new chain of models for avalanche hazard forecasting in French mountain ranges, including low altitude mountains, Proceedings of the International Snow Science Workshop Grenoble - Chamonix Mont-Blanc - 2013, 7-11 October, Grenoble, France, 162-166.

Sanpaolesi, Luca and Currie, D and Sims, P and Sacre, C and Stiefel, U and Lozza, S and Eiselt, B and Peckham, R and Solomos, G and Holand, I. and others. (1998). Scientific support activity in the field of structural stability of civil engineering works: snow loads. Final Report Phase I. Brussels: Commission of the European Communities. DGIII-D3.

---

## Author Response (AR1)

**Point by point reply to the Interactive comments of Anonymous Referees**

Dear Editors,

Thank you for giving us the opportunity to submit a revised draft of our manuscript «Non-stationary Extreme Value Analysis of Ground Snow Loads in the French Alps: a Comparison with Building Standards».

We thank the referee for their thorough reviews and for the numerous suggestions that helped us greatly improve the manuscript.
Following their comments, the main modifications are that we:

- added references for the validation of our data.
- validated qualitatively & quantitatively the selected models.
- modified the Abstract to better emphasize the content of the article
- changed the title of the manuscript.
- improved Figure 9 (and Section 5.3) with an additional axis illustrating the mean relative difference between return levels and French standards.
- corrected expression/syntax mistakes/typos across the document.
- generated all Figures in the pdf format instead of the jpg format.

Please find below, point by point, our answers to all the other suggestions. Anonymous Referee #1 suggestions are in red, while suggestions from Anonymous Referee #2 are in blue.

We hope that our revised manuscript will be found suitable for publication in "Natural Hazards and Earth System Sciences"

Yours sincerely,

On behalf of the co-authors
Erwan Le Roux

**2.** I missed some validation or references to validation of the GLS data. As mentioned by the authors, Safran has a number of biases. Crocus might be based on assumptions which are not always fulfilled and so the end product, GLS, might also suffer from a number of shortcomings.

*We added a paragraph in Sect. 2 emphasizing that the SAFRAN-Crocus reanalysis have been well validated.*

**3.** In addition, there is no validation of the GEV models, just the final selection among the models in Table 2. These are based on AIC and likelihood ratios. So the best model is selected. But do they fit well ? What if none of the models were really adequate (even the best one among them) ? Maybe some qq-plots analyses should be included.

*Quantile-Quantile (Q-Q) analysis is performed for all selected models. To apply this analysis to both stationary and non-stationary model, we rely on Coles (2001) that suggests 1) to transform the data to stationary Gumbel 2) to use a Q-Q plot analysis on the transformed data w.r.t. to a Gumbel distribution. Q-Q plots reveal that transformed data is well fitted by a stationary Gumbel distribution, hence that data is well fitted by the selected models.*

*Moreover, according to the comparative study of Abidin et al., (2012), the most powerful Goodness of Fit test for the Gumbel distribution is a combination of the Anderson-Darling test and the Maximum Likelihood Estimator. We apply this test on the transformed data, and found using Ali Saeb (2018) that we cannot reject the null hypothesis (samples generated from the Gumbel model) at the 5% significance level for 98% of the time series, justifying the good fit of our selected models.*

*We added these test results at the beginning of the Result section. In an Appendix section, we detail an explanation on the methodology of Coles (2001) and display Q-Q plots for the time series presented in section 2.*

**4.** Given the amount of literature, I found it a bit disappointing that no attempt was made to rely on models that make use of more data, not only annual maxima as mentioned in the discussion. For instance, the tail index is taken to be constant in view of the difficulty to estimate it. There are many ways around this, one of which is the so-called regional analysis.

*SAFRAN reanalysis are the result of a postprocessing of the meteorological observations at the massif scale and, as such, already represents "regionalized" data. In this context, it does not seem clear how a regional analysis could be performed.*

**5** The authors argue that the number of years of the GSL reanalysis is too short to attempt to use anything else than linear relationships in the non-stationary models. Nevertheless, they recognize that other extreme value approaches, such as peaks-overthreshold, can be apply to exploit more data (more than a maxima per year). This seems a bit contradictory. If the authors could show that the GEV models with linear non-stationarities fit well the data without too much uncertainty in the estimates, then it would alleviate this issue

*Our goal is to implement a clear comparison with French standards. For this reason, thus we prefer to rely on the Gumbel distribution & extensions of this distribution, which explains our choice to use Gumbel and GEV distributions.*

*Furthermore, the impact of the uncertainty in the estimates is already shown on our main figures (black bars on Figure 9). Despite that these uncertainty interval can sometimes be large, it does impact the main conclusions of this article. For instance, we would still have between 40 and 80% of massifs whose return levels in 2019 exceed French standards.*

**6.2. P.3 What is the spatial resolution of Safran ? Does GSL has the same spatial resolution ?**

*As explained on l.59, SAFRAN does not provide gridded data, it gives massif-level data. More precisely, as detailed in Nousu et al, (2019): " The principle of SAFRAN is to perform a spatialization of the available weather data in mountain ranges with so-called "massifs" of about 1000 km2 where meteorological conditions are assumed to depend only on altitude." In the Data section, we will add a sentence to make that point clear. Otherwise, Crocus snowpack model takes SAFRAN data as inputs to produce SWE (which we use to compute GSL), therefore yes, GSL data has the same spatial resolution as Safran.*

**6.2. P.11 last sentence : "... often above effective return levels " effective in what ways ? not sure what it means here**

*While classical stationary return levels do not depend on time, return levels are denoted as effective when they depend on time (Katz et al., 2002, Mondal et al., 2019) . To quote Katz et al., (2002): "[Effective design value] has an interpretation similar to that for an ordinary design value (i.e., the quantile corresponding to a specified return period), except that it varies depending on the time of year.".*

**6.3. -P.13 L.245-250 : " ... start the non-stationarity after the most likely year ", what is meant by most likely year ?**

*For each model with a linearity in some parameters of the distribution we could choose to start the linearity only after some starting year.*

*The most likely starting year is the year that gives the maximum likelihood for this linear model (Blanchet et al., 2016). However, in the end we decided not to use this approach. Therefore we remove this sentence altogether from the discussion Section to avoid confusing the reader with unnecessary details.*

**Specific comments**: The manuscript lacks a description of (i) error measures of GSL data used as basis of the extreme value statistics, and (ii) general remarks on the reanalysis used to provide that data. In particular it would be crucial to tell something about the BIAS or absolute errors of the yearly maximum GSL values. Otherwise provided uncertainty assessments are less valuable. Furthermore a general description of some aspects of the reanalysis is missing. How is GSL calculated for the massif scale? Is the 50-year GSL return level computed by your models valid for the whole massif just depending on altitude? The abstract of Vernay (2019) states also a dependency on aspect and slope. You should clarify if your results are valid for distinct elevations or elevation bands (as it is stated here and there). In the latter case you should explain, how GSL values are assigned to that band (see lines 59, 71 in your manuscript).

*The SAFRAN-Crocus reanalysis has been evaluated against various observation datasets, as reported in previous publications (Lafaysse et al., 2013, Vionnet et al., 2016, Revuelto et al., 2018, Vionnet et al. 2019). In most cases, the evaluation is carried out against in-situ snow depth observations and remote sensing snow cover information. For example, Vionnet et al., (2016) evaluated SAFRAN-Crocus snow depth data against 79 observed snow depth data in the French Alps for the 2010-2014 time period, with mean bias and standard error values of 18 cm and 37 cm, respectively. This corresponds to typical values for snow modelling systems applied in various regions on Earth. Because of lower data availability, evaluations against observed SWE values are less frequent than against snow depth data, although we note that Crocus has been shown to perform extremely well compared to other snow cover models, in terms of SWE, across many observation sites worldwide (Krinner et al., 2018) and SAFRAN-Crocus exhibits satisfying performance in terms of snow depth and SWE in the Pyrenees (Quéno et al., 2016), providing confidence, with respect to other existing datasets, in using this model chain for ground snow load (GSL) values. Further model evaluations, using additional datasets, are required to continue assessing and improving the quality of the model chain.*

*Furthermore, we highlight that we only used SAFRAN-Crocus reanalysis values on flat field, and we did not used simulations on slopes, hence it is not relevant to discuss the impact of slope and aspect on the results of this study.*

**Technical corrections:**
80: As maximum values are relevant in this study, the procedure of _removing the top annual maximum when considered exceptional_ should be shortly addressed. I can imagine that one can find information about that in the given reference, but this is in French…

*The procedure is as follows: «If the ratio of the largest load value to the characteristic load determined without the inclusion of that value is greater than 1.5 then the largest load value shall be treated as an exceptional value» (Sanpaolesi et al., 1998). This will be added to the revised version of the manuscript.*

84: What exactly do you mean with _relative change_? Relative to what? (see also line 48)

*We meant "relative change of 50-year return levels of GSL between 1960 and 2010". We will clarify it when necessary, and maybe refer to formula 4 (detailed expression).*

126: I wonder if these complex expressions are necessary to understand the content? If not you could remove them.

*We do not believe that the expression of the AIC is particularly complex. Most importantly, we think that this expression is necessary to understand the model selection, since the penalization of the log-likelihood by the number of fitted parameters clearly appears.*

219: Why of all things 1800 m? Is this because Vercors top heights are around 1800m?

*This is because French standards for extreme snow loads are defined from 200 m to 2000 m (Section 2). As we consider available altitudes between 200 m and 2000 m, only results obtained with reanalysis from 300 m to 1800 m are shown. However, the SAFRAN-Crocus reanalysis can provide results at higher elevation for the mountain areas peaking above 1800 m elevation.*

Figure 8. Top left panel: Do you have a clue, why the uncertainties at lower altitudes are larger than at higher altitudes? With respect to the smaller number of available reanalysis stations at higher altitudes, this should be inverted, as can be seen in all other panels.

*The reviewer must refer to the top-right panel (Vercors massif & Selected model) which is certainly different from the other panel.*

*Indeed, uncertainties usually grow larger with the altitude. Looking at similar plots to Figure 8 for all other massifs (not shown), this pattern is always seen for the left panels, i.e. with the stationary Gumbel model. However, for the right panels, i.e. for the selected model, 6 massifs out of 23 (Vercors, Ubaye, Oisans, Mercantour, Maurienne, Haut Var Haut Verdon) do not present this pattern, i.e. have larger uncertainties at lower altitudes. Some of these uncertainties might be due to variance in the estimated parameters. In particular, the shape parameter of the GEV distribution is known to be difficult to estimate. As shown in Figure 4, at 900 m the Vercors massif (most western massif) is colored in brown, meaning than its shape parameter roughly equals 0.3. This might explain the high uncertainty at 900 m in the top-right panel, as small changes around 0.3 can have large effect in the 50-year return level.*

257-258: You obtained the "same" results for time series with less than 10% of zero GSL values. Can you provide a similar number used by French standards for the decision to switch to a mixed discrete-continuous distribution?

*In the French standards, the mixed discrete-continuous distribution was considered for all time series, those with less than 10%, of zero GSL values, as well as those with more than 10%.*

299-300: This statement is unclear. I suggest to either remove it, or to provide more details. If you really would like to leave that here, you should provide at least a refer-ence for the European construction standards, and elaborate a little bit on those safety coefficients that might alter very widely according to country, professional, construction material, etc.

*We agree with the reviewer that this paragraph should be more detailed, and we did that in the revised manuscript. Concerning European standards (Sanpaolesi et al. 1998, page 32, equation 8), the design value for the structure equals the sum of i) the characteristic value of permanent action, i.e. self-weight, multiplied by a safety coefficient equal to 1.35 and ii) the characteristic value of variable action, i.e. roof snow load, multiplied by a safety coefficient equal to 1.5.*

Sanpaolesi, Luca and Currie, D and Sims, P and Sacre, C and Stiefel, U and Lozza, S and Eiselt, B and Peckham, R and Solomos, G and Holand, I. and others. (1998). 
[revised manuscript text omitted]

---

## Author Response (AR2)

**Point by point reply to the Interactive comments of Anonymous Referees**

Dear Editors,
Please find the final version of the manuscript attached,

We thank both Reviewer 1 and Reviewer 2 for their second thorough reviews and for their suggestions that helped us greatly improve the manuscript. Below, we answer to one question from Reviewer 1.

-> P.13 L.249 "Also, Figure 8 exemplifies the impact of accounting for decreasing trends in return levels." : I do not see why Fig.8 shows this.

I give the reason in the sentence following the cited sentence:
"Indeed, we observe that return levels from the stationary Gumbel model $M_0$ (left) are often larger than effective return levels in 2019 (last year of data) from the selected model $M_N$ (right)"
In other words, Figure 8 shows that the decreasing trends in return levels imply that effective return level in 2019 are smaller than stationary return levels. Indeed, we see on the Figure 8 that the green line on the left plot is often above the green line on the right plot (especially for the Beaufortain massif).

We hope that this answer will be found suitable.

Yours sincerely,

On behalf of the co-authors
Erwan Le Roux